# Periods of constant wind speed: How long do they last in the turbulent atmospheric boundary layer?

Daniela Moreno [1], Jan Friedrich [1], Matthias Wächter [1], Jörg Schwarte [2], and Joachim Peinke [1]

[1]Carl von Ossietzky Universität Oldenburg, School of Mathematics and Science, Institute of Physics. Oldenburg, Germany.
[2]Nordex Energy SE & Co.KG, Erich-Schlesinger-Straße 50, 18059 Rostock, Germany.

**Correspondence:** Daniela Moreno  (aura.daniela.moreno.mora@uni-oldenburg.de)

**Abstract.** We perform a statistical analysis of the occurrence of periods of constant wind speed in atmospheric turbulence. We hypothesize that such periods of constant wind speed are related to characteristic wind field structures which, when interacting with a wind turbine, may induce particular dynamical responses. Therefore, this study focuses on characterizing the constant wind speed periods in terms of their lengths and probability of occurrence. Atmospheric off-shore wind data are analyzed. Our findings reveal that long constant wind speed periods are an intrinsic feature of the marine atmospheric boundary layer. We confirm that the probability distribution of such periods of constant wind speeds follows a Pareto-like distribution admitting power law behavior for periods exceeding the large eddy turnover time. The power law characteristics depend on the local conditions and the precise definition of wind speed thresholds. A comparison to wind time series generated with standard synthetic wind models and to time series from ideal stationary turbulence suggests that these structures are not characteristics of small-scale turbulence but seem to be consequences of larger-scale structures of the atmospheric boundary layer and thus are multi-scale. Given the results, we show that the Continuous Time Random Walk model, as a non-standard wind model, can be adapted to generate time series of the wind speed whose statistics match the statistics of observed periods of constant wind speed.

## 1  Introduction

The estimation of the loads experienced by a wind turbine (WT) is fundamental for decision-making processes during the design phase of the various components of the machine, as well as for control strategies during its operation. Such estimation is performed through numerical modelling of the interaction between the WT and the incoming wind. Therefore, an accurate description of the wind within the atmospheric boundary layer (ABL) is essential for correctly calculating the loads acting on the WT. The International Electrotechnical Commission (IEC) has defined both, the widely-used standard parameters for the characterization of the atmospheric wind, and the models for generating synthetic wind fields used for numerical estimation of loads on the WT (IEC, 2019). These IEC standards consider the spectral properties and coherence of the velocity components of the wind. Nevertheless, such guidelines are designed to mimic the atmospheric wind computationally efficiently. As a result, some flow features in the ABL are neglected or simplified in the characterization of atmospheric measured data, as well as in the generation of the synthetic wind fields. Furthermore, during the past decades, new challenges in the design process of

WTs have emerged (Veers et al., 2019). On the one hand, trends in the design of modern WTs account for bigger rotor areas and less rigid structures (i.e. blades) to capture more energy from the available wind resources. On the other hand, the weight and material requirements of each component are being pushed to minimal levels. As a result, new WTs are becoming, in general, larger and less rigid. Therefore, some of the characteristics of the wind within the ABL that are not addressed in the IEC standard wind models might become relevant for extra loads previously neglected within the design of smaller and stiffer WTs.

Based on cooperative research with a WT manufacturer, we hypothesized that one of these features, disregarded by the IEC guidelines, is the periods of constant wind speed (CWS) in atmospheric flows. Such periods are defined as the intervals of time over which the magnitude of the wind speed remains almost constant within a certain range, limited by a threshold value. In the following, we first contextualize the periods of CWS within the general characterization of turbulent features. Afterwards, we discuss in which way such CWS structures may be relevant for a WT.

Concerning the CWS periods as a general feature of the wind, it should be mentioned that there are relevant and well-investigated turbulent quantities closely associated with our definition of CWS periods. That is the case of the persistence phenomenon, which characterizes how long the flow remains in a particular state before switching to another one. Persistence times can be inversely related to occurrence rates of extreme wind speeds or gusts. In this context, the exceedance statistics proposed by Rice (1944) have been applied for describing gusts as excursions at which certain thresholds of wind speed are exceeded (Kristensen et al., 1991; Young and Kristensen, 1992; Manshour et al., 2016). Another interpretation of persistence within turbulent flows is the so-called zero-crossing analysis. In this case, for a zero-mean signal, the waiting times between two successive crossings of its zero level are evaluated. Statistical properties of zero-crossings have been used to characterize intrinsic turbulent quantities such as the Taylor micro-scale (Narayanan et al., 1977; Sreenivasan et al., 1983; Kailasnath and Sreenivasan, 1993; Poggi and Katul, 2010) or the integral length scale (Mora and Obligado, 2020; Mazellier and Vassilicos, 2008). Analyses of zero-crossings of velocity and temperature fluctuations in atmospheric turbulent data have been discussed (Cava and Katul, 2009; Cava et al., 2012; Chamecki, 2013; Chowdhuri et al., 2020). To summarize, the above-mentioned investigations showed that the statistical characteristics of the persistence for experimental and atmospheric data exhibit a power law behavior up to a certain threshold, followed by log-normal or exponential cutoffs.

It is worth noting that even though the inter-arrival times of both, excursions and zero-crossings, refer to structures between particular turbulent states, they do not correspond to the periods of reduced turbulent amplitudes, in which we are interested. Further details of the differences between CWS periods and inter-arrival times between excursions and zero-crossings are shown in Appendix A. Nevertheless, the method and statistics of such persistent events are relevant to the discussion. Of special interest are self-similar, critical, or fractal features of turbulence that propose a power law behavior for the probability distribution of the time intervals with duration $T$, which can be formulated as $p(T) \propto T^{-\alpha}$ (in particular for the limit of large $T$). A characteristic feature of a power law distribution is the absence of an intrinsic scale, i.e., the probability of observing a realization larger than $\xi T$ is $\xi^{-\alpha+1}$ times the probability of observing a realization larger than $T$; independently of the value of $T$. The far-tail regime of many distributions occurring in complex systems is assumed to exhibit power law behavior (Laherrere

and Sornette, 1998). In the context of wind energy, for instance, a Pareto distribution has been tested as an extrapolation method to estimate extreme loads on a multi-megawatt wind turbine generator with a 1-month return period (Dimitrov, 2016).

Next, we discuss the potential relevance of an accurate description of the CWS periods for WT applications which is directly linked with the increasing size and flexibility of the WTs. In the simplest case, such periods of CWS should imply relatively quiescent operating conditions for a WT when the CWS structure occurs homogeneously in the rotor area. A more entangled case might occur when resonant or near-resonant dynamics appear for specific periods of CWS, over which the resonance can be strongly excited. In particular, for the larger WTs, the CWS periods may be restricted to a sub-area of the rotor plane. In this case, resonant dynamics exhibiting 3P oscillations may be amplified. Within this context, recent studies are devoted to interfaces between turbulent and non-turbulent states in atmospheric wind measured at typical WT heights (Neuhaus et al., 2024). Meanwhile, numerical and experimental investigations on the laminar-turbulent transition mechanisms on rotating wind turbine blades have shown changes in the transition characteristics over a single revolution, which affect the aerodynamic response of the WT (Lobo et al., 2023; Özçakmak et al., 2020).

As a last possible application for WTs, we want to mention that the statistical features of CWS periods may become of interest for probabilistic design methods. Although the methods proposed by the IEC for estimating WT loads are mostly deterministic (IEC, 2019), in recent years, probabilistic design methods have been introduced as surrogates for the design and load assessment of WTs (Abhinav et al., 2024; Kelma, 2024). Such probabilistic approaches account for more reliable estimations by considering the explicit calculation of the uncertainties from the operational conditions, aerodynamic models, materials, etc (Sørensen and Toft, 2010). Characteristics of the wind are then defined as stochastic variables within the probabilistic model. Accordingly, broader and more accurate statistical descriptions of the intrinsic features of the wind inside the ABL account for a reduction in the uncertainty of the estimated loads and responses of the WTs.

In this paper we focus on the periods of CWS as general features of turbulence; the discussion of possible impacts on a WT will be done only as side remarks. In particular, we characterize the statistics of periods of CWS (with a low level of turbulent fluctuations) from wind measurements in the ABL. In a preliminary investigation, the method for the assessment of such events from wind speed time series was presented and the first results on the characterization of the periods of CWS in terms of their duration and probability distributions were also reported (Moreno et al., 2022). Special attention within the characterization was given to the tails of the distributions, which describe extremely long periods. Interestingly, we found that the probability distribution for very long periods shows a power law decay $p(T) \propto T^{-\alpha}$. Furthermore, a comparison with wind data generated by an IEC standard model revealed that the model underestimates the frequency of occurrence of the extremely long CWS periods measured in the ABL. In this study, we aim to address whether the CWS periods are induced by specific orographic perturbations, whether they are laminar or low turbulent structures, and whether they are intrinsic features of a turbulent flow or, respectively, rather result from large-scale interactions within the ABL. To characterize the CWS periods we use data from offshore wind, as we expect to have less special orographic effects, compared to onshore data, and thus get more general insights of the CWS structure. This is also the motivation to compare the results with ideal turbulent data from a free jet

experiment. Furthermore, a stochastic wind field model for WT simulations is presented as a surrogate approach to incorporate the statistics of long CWS periods from turbulence in the ABL.

The paper is structured as follows: Sec. 2 retakes the method for measuring the periods and describes the atmospheric wind data to be analyzed. In Sec. 3, the results of the statistical characterization of the periods from the atmospheric data are shown. In Sec. 4, we compare the results from ABL data to those from two different data sets, i.e., IEC standard wind model and experimental ideal turbulence. In Sec. 5, we present our conclusions and potential future work.

## 2 Methodology and Data

### 2.1 Definition of a period of CWS

Following Moreno et al. (2022), a CWS period ($T_c$) is defined as the time over which the magnitude of the wind speed $u(t)$ exhibits low-amplitude fluctuations enclosed within certain thresholds. A period $T_c$ is depicted in Fig. 1. Over the length of $T_c$, the wind speed remains inside the constant speed range (CSR). The CSR is defined as $u_{t^*} \pm \varepsilon$, where $u_{t^*}$ is the reference speed value at $t = t^*$ and $\varepsilon$ is the maximum acceptable magnitude of the fluctuations around $u_{t^*}$. In Fig. 1, the horizontal red bars illustrate the thresholds that delineate the CSR. It should be noted that the CWS periods are not strictly laminar but periods with a smaller amplitude of turbulence; see also the spectral analysis in Sec. 3.

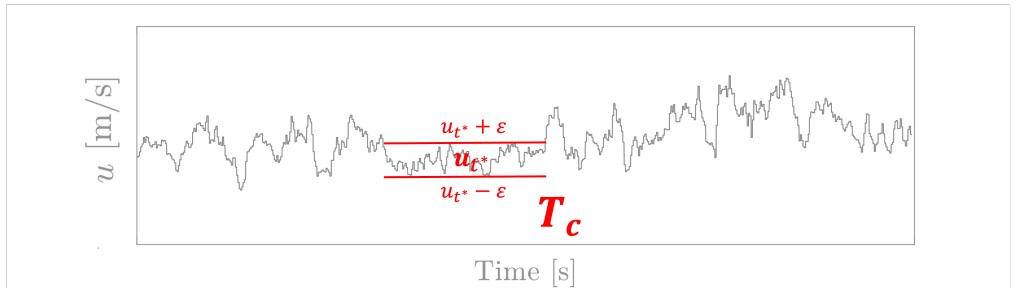

**Figure 1.** Schematic representation of a CWS period ($T_c$) measured from an exemplary wind speed time series $u(t)$. The constant speed range (CSR), $u_{t^*} \pm \varepsilon$, specifies the limits for the accepted level of turbulence within a period $T_c$. The CSR is depicted by the horizontal red bars.

In the following, the method for measuring the length of a period $T_c$ at a given time step $t^*$ is described in detail. The goal is to count the number $N$ of consecutive time steps, including $t^*$, for which their wind velocity $u(t)$ is contained inside the CSR. For that, the reference speed $u_{t^*} = u(t^*)$ and the corresponding CSR, $u_{t^*} \pm \varepsilon$, are defined. Next, the velocities at the time step $t^* + i$ for $i = (1, 2, 3..., \infty)$ are evaluated and counted. The counter $\tilde{N}^+$ for the evaluation of $u_{t^*+1} = u(t^* + i)$ is then defined as,

$$
\tilde{N}^+ = \begin{cases} \tilde{N}^+ = +1 & \text{if } (u_{t^*} - \varepsilon) \le u_{t^*+1} \le (u_{t^*} + \varepsilon) \\ \text{end} & \text{otherwise.} \end{cases} \tag{1}
$$

Note that only consecutive points are counted in $\tilde{N}^+$. The count is concluded once the value of $u(t^* + i)$ exceeds either the bottom or the top limits of the CSR. So far, only points in the forward direction (+) from $t^*$ are evaluated. The same algorithm is subsequently applied to counting the number of points $\tilde{N}^-$ in the backward direction from $t^*$. In this case, values of $i = (-1, -2, -3..., -\infty)$ are considered for evaluating $u_{t^*+1}$ in Eq. (1). Finally, the total number of consecutive points $N$ measured at $t^*$ results from the sum of $\tilde{N}^+$ and $\tilde{N}^-$, which are independently counted in their corresponding direction. The length of the period $T_c$ at $t^*$ is then obtained by multiplying the total $N$ by the size of the time step $\delta t$. A period $T_c$ is estimated for every time step in the time series $u(t)$. In the case of overlapping periods, only the longest-measured period is recorded. By doing so, a recounting of events is avoided.

In Moreno et al. (2022), the threshold $\varepsilon$ for fixing the CSR, $u_{t^*} \pm \varepsilon$, was randomly selected (e.g. 0.2 - 0.4 m/s), and the method described in Eq. (1) was applied over the actual measurements $u(t)$. However, limitations on the method appear when analyzing large data sets with very different mean wind speed $\bar{u}$ and standard deviation $\sigma_u$ calculated over shorter time windows (i.e. 10 minutes) with respect to the length of the sample. To introduce a systematic approach, in this paper, the threshold $\varepsilon$ is defined to be proportional to the standard deviation of the wind speed $\sigma_u$. Then, $\varepsilon$ for fixing $u_{t^*} \pm \varepsilon$ is calculated as,

$$
\varepsilon = A \, \sigma_u \tag{2}
$$

where $A$ is a factor, typically $A < 1$. The value of $A$ can be chosen depending on the particular application. In the case of a WT, $A$ might be related to the thresholds for the control system to operate within different turbulent regimes. In practice, such thresholds in the operating protocols are commonly defined as a function of the turbulence intensity $TI = \sigma_u/\bar{u}$. In Eq. (2) and through this document, we refer to $\bar{u}$ and $\sigma_u$ as the mean and standard deviation values, calculated over 10-minute periods unless a distinction is clearly stated.

## 2.2 Atmospheric wind data

Data from the offshore research platform FINO1 are investigated. We expect offshore wind to provide a better representation of undisturbed, or less disturbed conditions within the ABL compared to onshore data. Therefore, the possible effects of onshore orographic conditions on the CWS structures are diminished.

The FINO1 platform is located in the North Sea. Records of the wind speed $u(t)$ were taken by vertically aligned cup anemometers mounted at different heights $H$ (FINO). The data correspond to measurements from January to December 2007 with a sampling frequency of 1Hz. Measurements at heights $H = [30, 50, 70, 90]$ m above the mean sea level are considered. Wind speed records have been limited to those 10-minute periods with $\bar{u}$ between 3 and 25 m/s due to their relevance for WT operation. Values of $u(t)$ outside this range have been neglected. Moreover, to avoid disturbances from the met mast, data for wind

directions between 275 and 350° are not considered. As an overview of the complete data set, Fig. 2 shows the mean $\bar{u}$ and
standard deviation $\sigma_u$ calculated over individual 10-minute periods at $H = 90$ m.

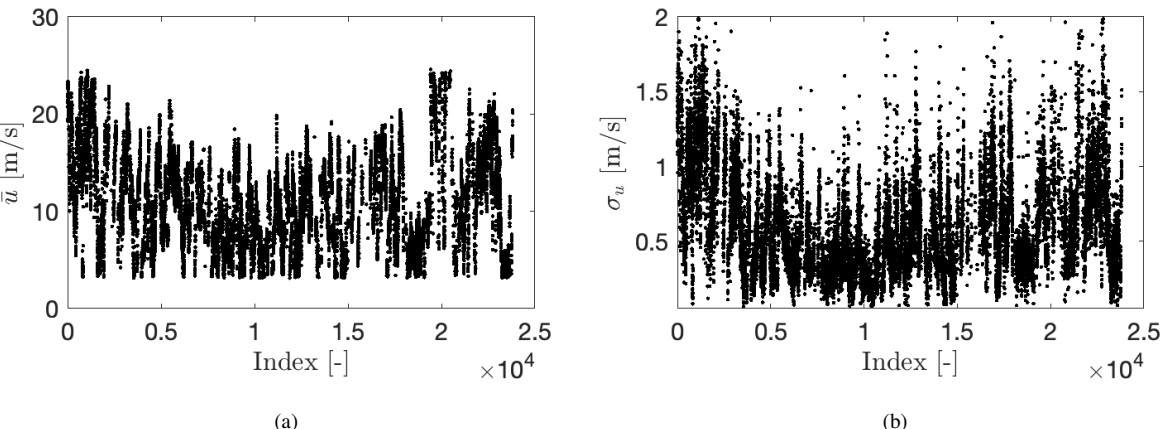

(a)                                                                              (b)

**Figure 2.** Wind velocity statistics of atmospheric FINO data at $H = 90$ m. (a) Mean wind speed $\bar{u}$. (b) Standard deviation $\sigma_u$. Each dot in
the plots corresponds to a calculated value over a single 10-minute period. The dots are chronologically ordered.

## 3  Statistics of $T_c$ for atmospheric turbulent data

### 3.1  Mean, standard deviation and maximum value of $T_c$

As a starting point on the statistical characterization of the measured CWS periods $T_c$, we discuss their mean duration ($\overline{T_c}$),
standard deviation ($\sigma_{T_c}$) and maximum value ($T_{c,max}$). We define $T_{c,max}$ as a representative value from a set of the longest
measured periods rather than the absolute and unique longest event. More details follow in Sec. 3.2. We compare the mentioned
statistics of $T_c$ at different heights $H$. A factor $A = 0.3$ is exemplary chosen for defining the threshold $\varepsilon = A\,\sigma_u$ for the CSR,
$u_{t^*} \pm \varepsilon$. The results are summarized in Table 1.

| $H$ [m] | **30** | **50** | **70** | **90** |
|---|---|---|---|---|
| $\overline{T_c}$ [s] | 3.6 | 3.6 | 3.7 | 3.6 |
| $\sigma_{T_c}$ [s] | 3.0 | 3.2 | 3.3 | 3.3 |
| $T_{c,max}$ [s] | 106 | 147 | 151 | 123 |

**Table 1.** Mean ($\overline{T_c}$), standard deviation ($\sigma_{T_c}$), and maximum length ($T_{c,max}$) of the calculated periods $T_c$ at different heights $H$. A factor
$A = 0.3$ is assumed for the estimation of $T_c$.

As a remark, special attention has to be devoted to the meaning of the statistical moments $\overline{T_c}$ and $\sigma_{T_c}$ calculated from the
data. In certain cases, as those presented in Moreno et al. (2022), the probability distribution $p(T_c)$ may lead to non-converging
moments, e.g., mean and variance. Further details are discussed in Appendices B and C. From the values in Table 1, comparable

$\overline{T_c} \approx 4\,\mathrm{s}$ and $\sigma_{T_c} \approx 3\,\mathrm{s}$ are obtained for the four heights $H$. More interestingly are the longest measured CWS periods $T_{c,max}$ at each height $H$. Periods with lengths up to $T_c \approx 40\,\sigma_{T_c}$ which correspond to more than $100\,\mathrm{s}$ are measured. The specific values of $\overline{T_c}$, $\sigma_{T_c}$ and $T_{c,max}$ are expected to be dependant on the specific local conditions due to surface interactions. Particularly, stronger differences in the lengths of CWS periods might arise under onshore conditions as observed by Kelly (2024) when analyzing coastal flow accelerations at different heights.

## 3.2 Probability density function of $T_c$

Next, in the statistical characterization of the CWS periods, the probability density functions (PDFs) $p(T_c)$ are discussed. Fig. 3 shows $p(T_c)$ for the data in Table 1 for different heights $H$. As mentioned before, we focus our attention on characterizing very long periods $T_c$. Therefore we concentrate on the tails of $p(T_c)$. For comparability, the values of $T_c$ are normalized by the longest measured period at each $H$; more precisely, we use a representative value $T_{c,max}$ of at least ten of the longest periods to become statistically more robust. The obtained values for $T_{c,max}$ are those summarized in Table 1.

The normalized PDFs $p(T_c)$ in Fig. 3 are presented in a log-log scale. In such a representation, a straight line reveals a power law behavior of the form $p(T_c) \propto T_c^{-\alpha}$ with $\alpha$ as the characteristic exponent. In Fig. 3, fitting power laws over the tails of the distributions are depicted by solid lines with the same color used for the dots at each $H$. This indicates that the PDFs of CWS periods $p(T_c)$ follow a Pareto-like distribution for large $T_c$ (Laherrere and Sornette, 1998). We emphasize that the power laws extend over more than one decade. The corresponding exponents $\alpha$ are calculated following the procedure proposed by Clauset et al. (2009) and described in Appendix D. The values of $\alpha$ are given in the legends of the figure.

The variation of the exponent $\alpha$ for all $H$ is enclosed within $\pm 6\%$. This shows that the decay $p(T_c) \propto T_c^{-\alpha}$ does not depend on the height. Moreover, since $\alpha \geq 3$ for all $H$, the second-order statistical moments of $T_c$ converge and the results presented in Table 1 provide meaningful information about the characteristics of the periods $T_c$ (see Appendices B and C).

The power law behavior observed in the distributions $p(T_c)$ for the offshore data shown in Fig. 3 agree with the obtained for the two onshore sites investigated by Moreno et al. (2022), as well as for analyses performed to data from the mast Wettermast Hamburg. This indicates that the CWS structures $T_c$ are not due to the specific orographic conditions, but rather represent general characteristics of the ABL. However, as the actual values of the statistics of CWS periods (i.e. $\overline{T_c}$, $\sigma_{T_c}$, $T_{c,max}$, $\alpha$) vary significantly between data sets, they should be exclusively considered for each location.

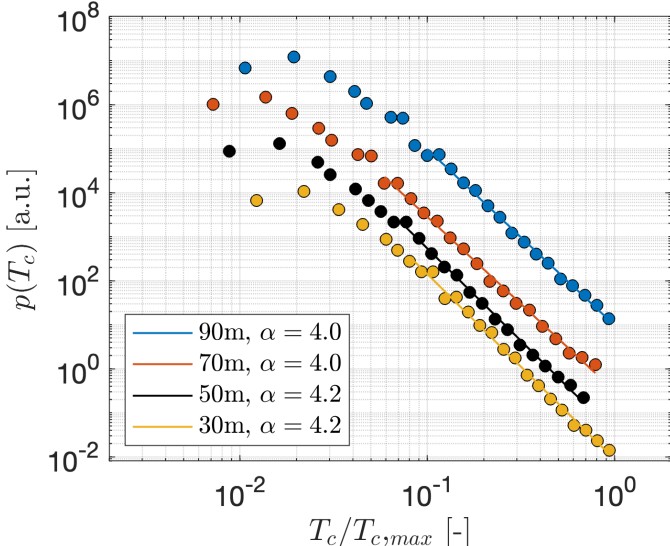

**Figure 3.** Normalized probability density functions $p(T_c/T_{c,max})$ for FINO data at different heights $H$. The dots illustrate the results from the FINO data. The solid lines show the power law decay fitting $\propto T_c^{-\alpha}$. The value $T_{c,max}$ for each height is defined as the bin centre containing at least ten of the largest measured periods after a binning process. The individual distributions are vertically shifted for better visualization.

### 3.3 Validity of the power law $p(T_c) \propto T_c^{-\alpha}$

To validate the universality of the power law distribution $p(T_c) \propto T_c^{-\alpha}$, we investigate the effect of the width of the CSR, $u_{t^*} \pm \varepsilon$. Different values of the factor $A$, as $\varepsilon = A\,\sigma_u$ are evaluated. The results of $\overline{T_c}$, $\sigma_{T_c}$, $T_{c,max}$, and $\alpha$ for $A = [0.2, 0.3, 0.5, 0.8]$ are summarized in Table 2. Respectively, Fig. 4 shows the normalized PDFs $p(T_c)$ in an analogue representation as shown previously in Fig. 3.

| $A$ [-] | 0.2 | 0.3 | 0.5 | 0.8 |
|---|---|---|---|---|
| $\overline{T_c}$ [s] | 3.0 | 3.6 | 5.3 | 9.4 |
| $\sigma_{T_c}$ [s] | 2.2 | 3.3 | 6.2 | 13.4 |
| $T_{c,max}$ [s] | 89 | 123 | 294 | 463 |
| $\alpha$ | 4.1 | 4.0 | 3.7 | 3.6 |

**Table 2.** Mean ($\overline{T_c}$), standard deviation ($\sigma_{T_c}$), maximum length ($T_{c,max}$), and exponent $\alpha$ of the calculated periods $T_c$ for different values of the factor $A$. FINO measurements at $H = 90\,\mathrm{m}$ are analyzed.

The tails of the PDFs in Fig. 4 show a clear power law decay $\propto T_c^{-\alpha}$ for all values of $A$. This confirms our hypothesis on the Pareto-like distributions of $p(T_c)$ for large $T_c$ already observed in Fig. 3. Interesting to note, is that the exponent $\alpha$ decreases with increasing width of the CSR, or the factor $A$.

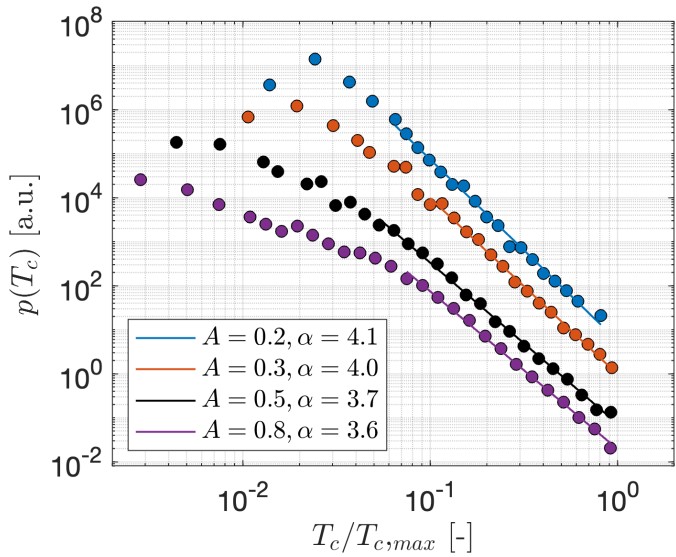

**Figure 4.** Normalized probability density functions $p(T_c/T_{c,max})$ for FINO data for different values of $A$. The power law fittings $\propto T_c^{-\alpha}$ are depicted by the solid lines. Measurements at $H = 90\,\mathrm{m}$ are considered. The value $T_{c,max}$ for each value of $A$ is defined as the bin center containing at least ten of the largest measured periods after a binning process. The individual distributions are vertically shifted for better visualization.

### 3.4 Power spectra of $u(t)$ during periods $T_c$

Further in the characterization of the CWS periods, the spectral features of the wind speed $u(t)$ during the CWS periods $T_c$ address the question of whether the wind speed is strictly laminar, or rather turbulent with a low degree of turbulence. The turbulent nature of $u(t)$ is now verified by the power spectra, shown in Fig. 5. The spectra E(f) are calculated from the extracted time series of $u(t)$ during CWS periods larger than $10\,\mathrm{s}$. The time series $u(t)$ are normalized by the standard deviation $\sigma_u$ of their corresponding 10-minute period. A time window of roughly five days was considered for extracting the definite time series $u(t)$ during $T_c > 10\,\mathrm{s}$. A decay of the form E(f) $\propto \mathrm{f}^{-5/3}$ is obtained for all heights $H$. Accordingly, the wind data embedded along the periods $T_c$ are not laminar flow sections but periods of turbulence with smaller amplitudes.

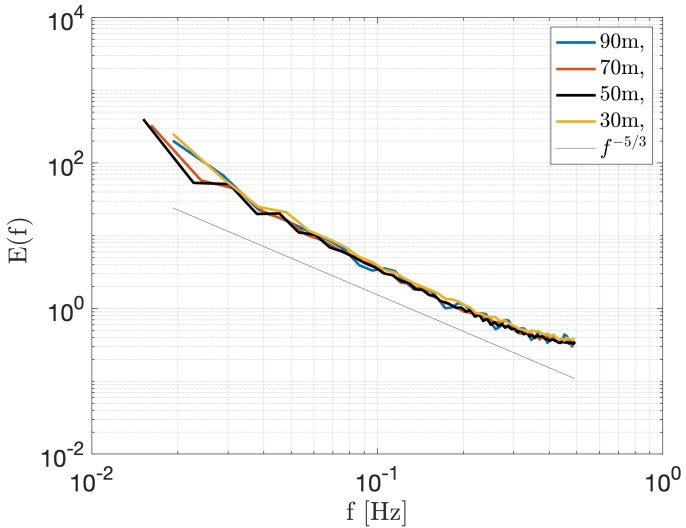

**Figure 5.** Power spectra E(f) of normalized wind speed $u(t)$ during the measured periods $T_c$ at different heights $H$. The grey solid line shows a decay $E(f) \propto f^{-5/3}$. The spectra are calculated for each period $T_c$ and then averaged over all periods. A time window of roughly five days was considered for extracting the definite time series $u(t)$ during $T_c > 10\,\mathrm{s}$.

## 4 Comparison to pure turbulent and synthetic wind data

### 4.1 Experimental wind-tunnel turbulence and IEC-standard Gaussian Kaimal

In order to investigate whether the CWS periods are typical features of turbulent flow or are special features of the ABL, we investigate the statistics of the CWS periods $T_c$ from experimental wind-tunnel turbulent data, as well as from synthetic data. The experimental data, 'Lab', was measured by Renner et al. (2001) in the central region of a free jet, which is approximately stationary, homogeneous, and isotropic. The synthetic data, 'Kaimal', corresponds to IEC-standard wind data based on the well-known Kaimal model, with normally-distributed amplitudes (Kaimal et al., 1972). The 'Kaimal' data are generated by

the NREL Turbsim package (Jonkman, 2016). Details about the parameters and characteristics of the two additional data sets, 'Lab' and 'Kaimal', are given in Appendix E.

The analysis of the CWS periods from FINO and Kaimal could be easily compared as the wind data sets $u(t)$ have comparable IEC-standard characteristics in terms of mean wind speed, standard deviation, sampling frequency, and integral length scale. However, such a match of parameters to atmospheric data is not possible with the experimental Lab data. To work out the

205 intrinsic features of the periods of CWS from these different data, we used two different approaches for normalizing the calculated $T_c$.

Firstly, the normalization is done by $T_{c,max}$ analogous as in Fig. 3 and Fig. 4. The resulting normalized PDFs $p(T_c)$ for the three wind data sets: FINO, Kaimal, and Lab are shown in Fig. 6. For its interpretation, it is important to remark that the

number of data points, given by the sampling rate and measured time, determines the lowest probability that can be resolved

within the PDF. Accordingly, the minimum value of $p(T_c)$ for Kaimal data in Fig. 6 is explained by fewer data in the sample. Oppositely, the high probability $p(T_c)$ of shorter periods for Lab data is explained by a much higher sampling of the data.

Clear different PDFs are observed for the three data sets in Fig. 6. The most prominent power law $p(T_c) \propto T_c^{-\alpha}$ is found for the FINO data with a smaller exponent $\alpha$ or more heavy-tailed probabilities. For the Kaimal and Lab data, a power law can be put into question. We show nevertheless power laws as a reference for comparison between the three data sets. Interestingly,

the range of periods $T_c$ for which the power law holds for the FINO data extends over a decade, at least from $T_c = 0.1\,T_{c,max}$ to $T_c = T_{c,max}$. In contrast, the power law range for Kaimal and Lab spreads only from $T_c = 0.3\,T_{c,max}$ to $T_c = T_{c,max}$.

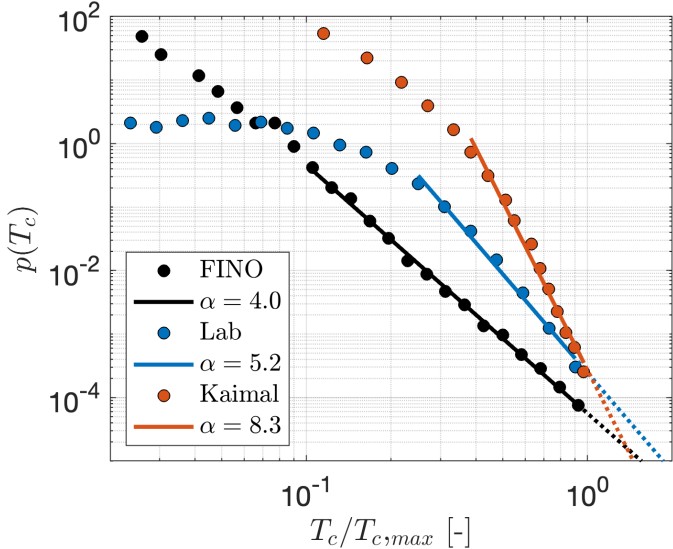

**Figure 6.** Normalized probability density functions $p(T_c/T_{c,max})$ for FINO, Kaimal and Lab data sets. The power law fittings $\propto T_c^{-\alpha}$ are depicted by the solid lines. The value $T_{c,max}$ for each data set is defined after a binning process as the center of a bin containing at least ten of the largest measured periods. Measurements at $H = 90\,\mathrm{m}$ are considered for FINO. The threshold $\varepsilon = A\,\sigma_u$ for the CSR is calculated with $A = 0.3$. The values of $\sigma_u$ for Kaimal and Lab are $0.58\,\mathrm{m/s}$ and $0.38\,\mathrm{m/s}$, respectively. In this particular case, as both data sets are expected to be steady, the standard deviation $\sigma_u$ is calculated over the length of the time series.

The normalization by $T_{c,max}$ shown in Fig. 6 does not provide any information regarding the magnitude of the CWS periods $T_c$. Therefore, a comparison of absolute values $T_c$ between the three data sets remains inconclusive. Accordingly, we chose a second approach for normalizing the CWS periods so that their lengths are related to the intrinsic lengths of the flow. The

220 integral length scale $L_{int}$ is a measure of the longest correlations. For ideal turbulence, structures that are significantly larger than $L_{int}$ are not to be expected. For meteorological wind data, the problem arises that at lower frequencies no white noise (i.e. zero correlation) is present so that larger structures than $L_{int}$ are expected (Sim et al., 2023; Larsén et al., 2016). Thus we now normalize the periods $T_c$ by the so-called large eddy turnover time $T_{int} = L_{int}/\bar{u}$ (Monin and Yaglom, 2007), where $\bar{u}$ is

calculated over the full time series for Kaimal and Lab data. The resulting PDFs $p(T_c)$ after the second normalization approach
are shown in Fig. 7.

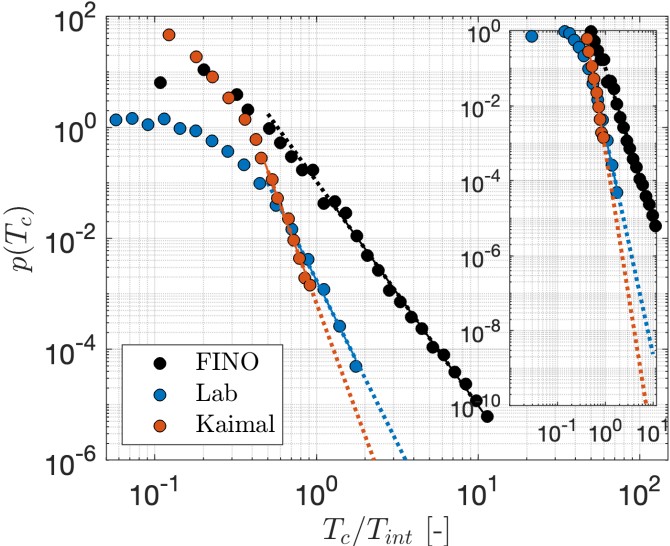

**Figure 7.** Normalized $p(T_c/T_{int})$ probability density functions for FINO, Kaimal and Lab data sets. The values of $T_{int}$ are, respectively, $17\,\mathrm{s}$ and $0.029\,\mathrm{s}$ for Kaimal and Lab(Fuchs et al., 2022). For FINO, $T_{int}$ is considered to be $10\,\mathrm{s}$ as a representative value of the atmospheric data. The power law fittings $\propto T_c^{-\alpha}$ are depicted by the solid lines. The dotted lines show the power law fittings extended over a range of $T_c$ larger than the range used for calculating the fitting parameters.

Fig. 7 shows that the FINO data have significantly longer CWS periods $T_c$. It is observed that the maximal CWS event of the data from the Gaussian Kaimal model, $T_c \approx T_{int}$, is around 100 times more frequent for the FINO data than for the other two data sets. Assuming the extended power law tails for Kaimal and Lab depicted by the dotted lines, and better visualized in the zoomed plot, a period $T_c \approx 10\,T_{int}$ would be around $10^4$ times less probable in the Kaimal and Lab data compared to the
measured FINO data. From the 1-year FINO data, we measured 15 events $T_c \approx 10\,T_{int}$ (with $p(T_c) = 5.7\mathrm{x}10^{-6}$). Roughly, it means an observation $T_c \approx 10\,T_{int}$ every 24 days. Under the IEC Kaimal's Gaussian assumption, this event will appear once every $66\mathrm{x}10^4$ days or 1808 years.

Furthermore, we calculate the standard deviation of the periods $\sigma_{T_c}$ in units of integral lengths $L_{int}$. The resulting values are $\sigma_{T_c,Lab} = 0.12\,L_{int}$, $\sigma_{T_c,Kaimal} = 0.09\,L_{int}$, and $\sigma_{T_c,FINO} = 0.31\,L_{int}$. The estimated values of $\sigma_{T_c}$ show in another way
that FINO data tend to remarkably longer periods, compared to Kaimal and Lab.

**4.2  CTRW wind model**

We have shown results on the distributions of CWS periods $p(T_c)$ in the ABL and their underestimation by the IEC-standard Gaussian Kaimal wind model. Consequently, we finally show how the observed features of the atmospheric turbulent data can

be included in a numeric wind field model. As a surrogate for the IEC-standard Kaimal model, we investigate non-standard

wind velocity time series generated by the Continuous Time Random Walk (CTRW) model (Kleinhans, 2008; Ehrich, 2022; Schwarz et al., 2019; Mücke et al., 2011). The CTRW model generates either Gaussian 'CTRW-G' (-G as an abbreviation of Gaussian) or non-Gaussian (-NG as for non-Gaussian wind velocity time series. For the 'CTRW-G', the statistics of $u(t)$ are entirely Gaussian. On the contrary, the statistics of $u(t)$ for the 'CTRW-NG' deviate from Gaussianity towards distributions with so-called heavy tails or higher probabilities of rare or extreme events.

The CTRW model is based on a skewed Lévy-distributed stochastic process, parameterized by the characteristic exponent $\alpha_L$. The stochastic process defines a time transformation from the intrinsic scale of the model $s$ to the physical time $t$. Such time-scaling transformation allows the generation of non-Gaussian time series $u(t)$. The characteristic exponent $\alpha_L$, with $0 < \alpha_L \leq 1$, specifies the asymptotic behaviour of the skewed Lévy distribution. For $\alpha_L$=1 the resulting process $u(t)$ is entirely Gaussian. Values of $\alpha_L \to 0$ generate processes with more pronounced non-Gaussian characteristics. In this case, non-Gaussianity is

related to extremely long waiting times between two successive time steps $s$. A very long waiting time in $u(s)$ would then be translated into a period over which the process $u(t)$ remains constant.

Fig. 8 shows an excerpt of $u(t)$ for a Gaussian CTRW-G and a non-Gaussian CTRW-NG realizations. Respectively, values of $\alpha_L = 1$ and $\alpha_L = 0.9$ are considered. Along the interval between $t = 875\,\text{s}$ and $t = 895\,\text{s}$, a period of almost constant wind speed is observed for the CTRW-NG. For better visualization, a zoomed version of the time series is presented in the sub-figure

in the right-bottom corner. Such a structure of the wind, indicated by the horizontal blue line, agrees with our definition of a CWS period $T_c$. The observed small fluctuations within the CWS period result from the interpolation process between the intrinsic and the physical times $s \to t$ (Ehrich, 2022).

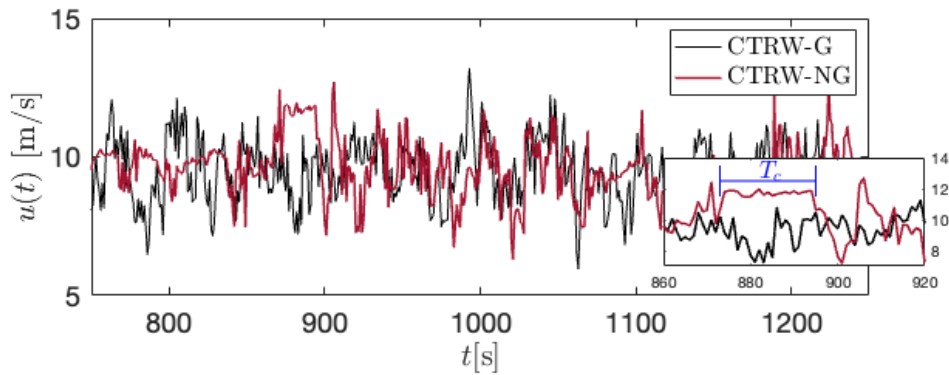

**Figure 8.** Excerpt of the wind speed time series $u(t)$ for CTRW-G and CTRW-NG. A CWS period $T_c$ is visible between 875 and 895s for the CTRW-NG. The exponent for the Lévy distribution of the CTRW-NG is $\alpha_L = 0.9$.

The fundamentals of the CTRW model as well as further details on the method for achieving such non-Gaussian features are given in Appendix F. The parameters for generating the time series are provided in Appendix E.

Fig. 9(a) shows the PDFs $p(T_c)$ for the CTRW realizations, and the FINO data. The individual distributions are vertically shifted for better visualization. The dotted lines show the Gaussian distributions with the mean and standard deviation of the corresponding $p(T_c)$. The grey-shadowed area illustrates the range of the decays of $p(T_c)$ or slopes $\alpha$, enclosed by CTRW-G ($\triangle$) with $\alpha_L = 1$, and CTRW-NG ($\square$) with $\alpha_L = 0.9$. The distribution of the CTRW-NG realization shows an overestimation, compared to the FINO data, of the deviation from Gaussianity towards a higher probability of very long-duration periods $T_c$. This deviation is visible from $T_c \approx 0.3\, T_{c,max}$. On the contrary, the decay of the CTRW-G is much more pronounced and the divergence from the Gaussian distribution is visible only for events $T_c > 0.6\, T_{c,max}$. A third realization, 'CTRW-NG∗' ($\bullet$) with $\alpha_L = 0.995$ is included. The resulting $p(T_c)$ distribution for CTRW-NG* shows a better agreement with the FINO data. Both distributions, FINO and CTRW-NG*, lie inside the grey shadowed area depicting the slopes enclosed between the Gaussian CTRW-G and extremely non-Gaussian CTRW-NG.

Fig. 9(b) shows the resulting exponents $\alpha$ from the decay $p(T_c) \propto T_c^{-\alpha}$, against the exponent $\alpha_L$ from the Lévy distribution of the CTRW model. The dotted horizontal line depicts the value of $\alpha$ for FINO in Fig. 9(a). As observed, by tuning the $\alpha_L$ parameter of the CTRW model, non-Gaussian realizations of $u(t)$ can reproduce the statistics of $p(T_c)$ from turbulent wind in the ABL. Since the resulting distributions $p(T_c)$ are considerably sensitive to the Lévy exponent $\alpha_L$, a careful selection of the exponent is required.

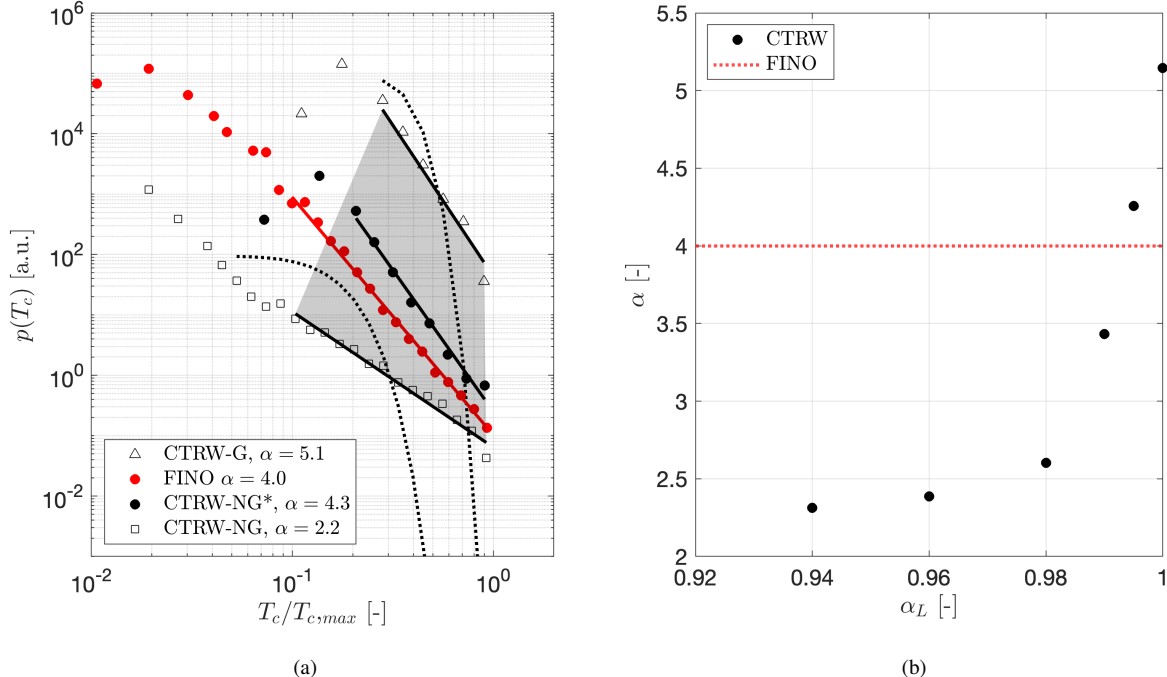

**Figure 9.** (a) Normalized probability density functions $p(T_c/T_{c,max})$ for the CTRW-G, CTRW-NG, CTRW-NG*, and the FINO data. The grey area depicts the range of the slopes covered between CTRW-G and CTRW-NG. Measurements at $H = 90\,\mathrm{m}$ are considered for FINO. The individual distributions are shifted vertically for better visualization. Dotted lines depict Gaussian distributions. (b) Power law exponents $\alpha$ from $p(T_c) \propto T_c^{-\alpha}$ as a function of the characteristic exponent $\alpha_L$ from the Lévy distribution of the CTRW model. The horizontal red line depicts the value of $\alpha$ for FINO data shown in (a).

## 5 Conclusions and Outlook

We present measurements of the CWS periods ($T_c$) (periods with turbulence of a reduced amplitude) from offshore wind data within the ABL. It is shown that the probability distributions $p(T_c)$ for offshore data exhibit a power law decay $p(T_c) \propto T_c^{-\alpha}$ for very long events (i.e. hundreds of seconds). This agrees with Moreno et al. (2022), where preliminary results from onshore cases were reported. However, significant differences in the values of the exponent $\alpha$ between offshore and onshore conditions suggest that the lengths of $T_c$ are indeed influenced by interactions with the surroundings. Therefore, the estimated statistics of $T_c$ must be considered locally for the specific location of interest. Given that offshore conditions maintain a more unperturbed ABL compared to onshore, we demonstrated that the periods $T_c$ are intrinsic features of the ABL rather than resulting structures originated by specific external factors (i.e. mountains, obstacles). Moreover, the exponent $\alpha$ seems to be quite independent of the height but changes significantly with the threshold $\varepsilon$. Less pronounced decays of $p(T_c)$ are obtained with wider thresholds for considering the wind speed as constant. We found examples of $T_c$ significantly larger than $100\,\mathrm{s}$, which correspond to

spatially extended structures over sizes larger than 1 km, using Taylor's hypothesis of frozen turbulence. Such large structures in turbulent wind may be related to the so-called and currently very discussed 'turbulent superstructures' (Pandey et al., 2018; Krug et al., 2020; Käufer et al., 2023).

Based on the spectral properties, we proved the turbulent nature of the wind speed $u(t)$ during the CWS periods $T_c$. This relates our results to the case of the turbulent-turbulent interfaces (Kankanwadi and Buxton, 2022). However, the statistics of $T_c$ are deviate significantly when comparing different turbulent data. Results from experimental homogeneous isotropic turbulence data suggest that the nature of the periods $T_c$ is attributed to special structures developing in the wind inside the ABL. It is still an open question whether they are caused by special effects of the small-scale turbulence (such as turbulence with or without shear) or whether they are indeed consequences of larger-scale interactions of the atmospheric boundary layer, such as phenomena related to the spectral gap (Larsén et al., 2016).

The frequency of very long events $T_c$ in the ABL is significantly underestimated by the Gaussian assumptions in the IEC models. Therefore, the need for an improved wind model is justified. The Continuous Time Random Walk (CTRW) model, with its characteristic time mapping (see Appendix F), is particularly suitable for the incorporation of the periods $T_c$ measured from the atmospheric turbulent wind. By tuning the exponent of the intrinsic Lévy distribution, different statistics of very long CWS periods can be obtained. This surrogate wind model represents an improvement towards more realistic atmospheric wind fields for numerical simulations. Consequently, responses of the WT interacting with such disregarded structures on the wind might be better predicted.

From an engineering perspective, very long CWS periods might be undesirable for the operation of WTs if phenomena such as resonance or critical loading are induced. On the other hand, they also might be beneficial if conditions such as constant power production are achieved. Further research is needed on the detailed effects of CWS periods on loads by investigating specific WT models.

A very long CWS period might have an increased impact on a WT depending on its spatial location in the plane of the rotor. The effect of such an event happening in the outer region of the rotor plane might be higher compared to the case when it reaches the turbine at the region near the hub. Accordingly, preliminary investigations (detailed in Appendix G) suggest that the periods of CWS show a tendency to be localized at different measurement heights, and therefore, may become of particular interest for turbines with larger diameters. Future work has to be devoted to assessing the relevance of the empirically observed power law behavior of periods of CWS on turbine loading. For that, the complete statistical parametrization of periods of CWS, in both time and spatial domains, should be assessed and improved for the synthetic wind field models such as the here proposed CTRW model (Kleinhans, 2008), the recently introduced Time-mapped Mann model (Yassin et al., 2023) which can generate long waiting times of $u(t)$ as in the CTRW model, or the Superstatistical model (Friedrich et al., 2021, 2022) that follows the K62 model of turbulence. Another interesting aspect for future work would be the statistical analysis of CWS periods from weather-modelled data (e.g. ECMVF, WRF models). The results would reveal whether such larger-scale models can reproduce the CWS structures within atmospheric forecasting.

*Code availability.* The code of the algorithm described through Sec. 2.1 for measuring the CWS periods from wind speed data can be

provided upon request.

*Data availability.* The FINO and Lab measurements, as well as the generated Kaimal and CTRW time series can be obtained upon request.

## Appendix A:  CWS periods vs persistence events

In the Introduction (Sec. 1) we referred to the inter-arrival times of excursions and zero-crossings as two general turbulent

characteristics within the context of persistence phenomena. Those inter-arrival times might wrongly be assumed as intrinsi-

cally related to our periods of CWS. As shown in Fig. G1, fundamental differences arise when comparing the three events

within a turbulent signal. In the upper part of Fig. G1 a zero-mean and normalized-by-standard-deviation signal $(u(t) - \bar{u})/\sigma_u$

is plotted. The thresholds $\pm\mathcal{U}$ are fixed for considering the excursions of the signal. The blue area depicts the range contained

inside these thresholds. The excursion events are depicted by blue crosses. Similarly, the zero-crossings are depicted by red

crosses. The grey rectangles depict the measured CWS periods $T_c > 10\,\mathrm{s}$. We assume $\varepsilon = 0.3$ for measuring $T_c$.

At the bottom part of Fig. G1 the length of selected inter-arrival times between the excursion and zero-crossing events, and the

periods $T_c$ are plotted. The lines follow the color code on the top plot. The selected inter-arrival times are only those longer

than 10s, as was assumed for the periods $T_c$. As an additional criterion for the excursions (blue lines), only inter-arrival times

between successive excursions at the upper limit $+\mathcal{U}$ are considered (i.e. events at $-\mathcal{U}$ are neglected). Note that if all the inter-

arrival times were plotted without any distinction, then the individual blue and red lines will append to each other covering the

entire length of the time series.

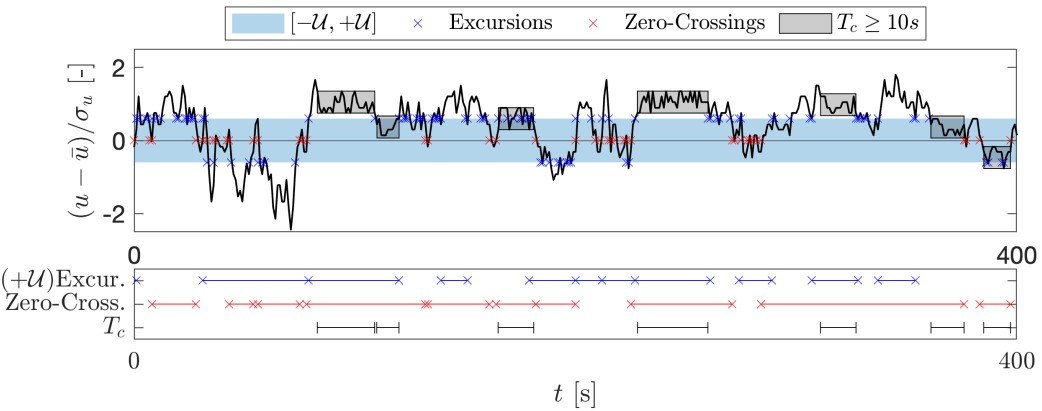

**Figure A1.** Illustration of excursions, zero-crossings and CWS periods ($T_c$). On top, a normalized signal $(u(t) - \bar{u})/\sigma_u$. The blue crosses depict the excursions, considering $\pm\mathcal{U}$ as thresholds. The red crosses correspond to the zero-crossings. The grey rectangles mark periods of CWS $T_c > 10\,\text{s}$. The blue and red lines in the bottom plot depict a selection of the resulting inter-arrival times for the excursion measured at the upper limit $+\mathcal{U}$, and the zero-crossings, respectively. Only the inter-arrival periods longer than $10\,\text{s}$ are shown. For comparison, the periods $T_c > 10\,\text{s}$ are re-plotted as black lines.

As observed, there is no direct correlation between the occurrence or the length of the CWS periods and the inter-arrival times; neither between excursions nor between zero-crossings. A CWS period might enclose several inter-arrival times, as well as several CWS structures might be embedded inside an interval between consecutive zero-crossings or excursions.

### Appendix B: Power law distributions

A general quantity $x$ with a probability distribution $p(x)$ follows a so-called *power law* if

$$p(x) = C\,x^{-\alpha} \tag{B1}$$

for $x \geq x_{min}$ with the characteristic exponent $\alpha$ and a constant $C = e^c$. The minimum value $x_{min}$ holds for the lowest limit of the power law. The exponent $\alpha > 1$, otherwise $\int_0^\infty x^k p(x)$ does not converge. The estimation of $\alpha$ from empirical data has been extensively discussed in the analysis of the distributions of a very wide range of applications (Newman, 2005; Clauset

et al., 2009). Since Eq. (B1) is equivalent to $\ln p(x) = -\alpha \ln x + c$, the most simple approach for the calculation of $\alpha$ comes from a linear regression on the log-log plot of the histogram of $x$. However, this procedure introduces significant errors due to the binning of the data and the resulting distributions. Such distributions are usually dominated by a few bins at lower values of $x$ with very high values of $p(x)$, and several bins at the higher range of $x$ with very low probabilities $p(x)$ (Newman, 2005; Dorval, 2008). Instead of such a linear regression, a logarithmic binning process of the data is recommended. Within this

approach, the histogram of $x$ is constructed for $k$ number of bins with variable width. More specifically, the bin edges $B$ are

proportional to successive powers of a constant $a$. Then,

$$B = (b_1, b_2, ..., b_{k+1}) = x_{c,min}(a^0, a^1, ..., a^k) \tag{B2}$$

where $b_1 > 0$, $k > 1$ and $x_{min}$ is the minimum value of $x$ for considering the power law behaviour. Thus, the $i_{th}$ bin encloses the interval $[b_i, b_{i+1})$ and the larger edge of the $k_{th}$ is assumed to be $+\infty$.

The value of the lower bound $x_{min}$ affects the estimation of the exponent $\alpha$ in $p(x) \propto x^{-\alpha}$. Analogously, for binned data, $b_{min}$ is defined as the minimum bin taken into consideration for the calculation of $\alpha$. We follow the algorithm proposed by Clauset et al. (2009), and Virkar and Clauset (2014) for choosing $b_{min}$ from binned empirical data. This method is based on a Kolmogorov-Smirnov (KS) statistic test (Massey, 1951) for minimizing the distance between the distributions of the fitted model $P(b|\alpha, b_{min})$ and the empirical model $S(b)$ above $b_{min}$. Then, the optimized value of $b_{min}^*$ minimizes

$$D = \max_{b \geq b_{min}} |S(b) - P(b|\alpha, b_{min})|. \tag{B3}$$

Further details about the method for calculating $b_{min}$ and $\alpha$ are provided in Appendix D.

## Appendix C: Statistical moments of power laws

A power law distribution of a continuous variable $x$ is defined in Eq. (B1), where $\alpha > 1$ is the power law exponent, $C$ is a normalization constant, and $x_{min}$ is the minimum value at which the power law holds. Then, the $k^{th}$ statistical moment of a power law distribution $p(x) = C x^{-\alpha}$ is given by,

$$\langle x^k \rangle = \int_0^\infty x^k p(x) \mathrm{d}x = \underbrace{\int_0^{x_{min}} x^k p(x) \mathrm{d}x}_{:=\tilde{A}} + \int_{x_{min}}^\infty x^k p(x) \mathrm{d}x = \tilde{A} + \frac{C}{k+1-\alpha} \left[ x^{-\alpha+k+1} \right]_{x_{min}}^\infty. \tag{C1}$$

Then, a quantity $x$ with $p(x) \propto x^{-\alpha}$ may have divergent moments. Extensively, its general $k^{th}$ moment exists only if $k < \alpha - 1$. The mean value of $p(x)$, or $\langle x^1 \rangle$, becomes infinite for $\alpha \leq 2$. Furthermore, if $\alpha \leq 3$, $p(x)$ has no finite variance, $\langle x^2 \rangle$. In such a case, $x$ can take values of $\bar{x} \pm \infty$. Many phenomena, varying from biological to economical, are characterized by such so-called critical distributions. A few examples are the frequency of use of words, the income among individuals, and the magnitude of earthquakes (Newman, 2005; Marquet et al., 2005; Powers, 1998).

## Appendix D: Estimation of $b_{min}$

Here we describe the method for estimating the minimum bin $b_{min}$, introduced in Appendix B, above which the power law $p(T_c) \propto T_c^{-\alpha}$ is valid. The method was proposed by Virkar and Clauset (2014).

For each possible $b_{min} \in (b_1, b_2, ..., b_{k/2})$,

1. Calculate the cumulative binned empirical distribution $S(b)$ for bins $b \geq b_{min}$.

2. Estimate the characteristic exponent $\tilde{\alpha}$ considering $b \geq b_{min}$.

3. Calculate the cumulative density function (CDF) for $P(b|\tilde{\alpha}, b_{min})$ of the binned power law.

4. Calculate the Kolmogorov-Smirnov(KS) test statistic $D$ defined in Eq. (B3).

5. Select the optimal value $b_{min}^*$ as the value of $b_{min}$ with the minimum test statistic $D$.

The bins $b$ are defined according to Eq. (B2). For the estimation of $\tilde{\alpha}$ in step (2.), a least-squares linear regression method is considered.

## Appendix E:  Further details of wind tunnel experimental and synthetic IEC standard wind data

- Kaimal: The data set contains $4\mathrm{x}10^5$ data points with a frequency of $1\,\mathrm{Hz}$. The implementation in TurbSim (Jonkman,
2016) of the Kaimal spectrum for the longitudinal component $u$ of the wind follows,

$$S_u = \frac{4\sigma_u^2 L_u/\bar{u}_H}{(1 + 6\,f\,L_u/\bar{u}_H)^{5/3}} \tag{E1}$$

where $\sigma_u$ is the standard deviation, $\bar{u}_H$ is the mean at hub height, and $f$ is the frequency. The integral scale $L_u$ is defined as $L_u = 8.10\Lambda_u$, with $\Lambda_u$ being the turbulence scale. $\Lambda_u$ is calculated as $\Lambda_u = 0.7\,(\min\{30\mathrm{m}, H_H\})$, where $H_H$ is the hub height. The parameters are chosen to be comparable to the averaged values of FINO data (see Sec. 2.2). We assume
the hub height $H_H = 90\mathrm{m}$, the mean wind speed $\bar{u}_H$ is $10\,\mathrm{m/s}$ and the standard deviation $\sigma_u$ is $0.58\,\mathrm{m/s}$. Then, the integral length scale is set to $170\,\mathrm{m}$.

- CTRW: Both realizations, 'CTRW-G' and 'CTRW-NG', have $4\mathrm{x}10^5$ data points with a frequency of $1\,\mathrm{Hz}$. The mean wind speed and standard deviation are $9.5\,\mathrm{m/s}$ and $1.1\,\mathrm{m/s}$ for both cases. Extended parameters for the model are $\omega_c = 1.8\,\mathrm{Hz}$; $\alpha_L = [0.9, 1]$, and $\tilde{c} = 350$. Details on the definition of the parameters are given in Appendix F and by Ehrich (2022).
The values of the parameters are chosen to generate data comparable to FINO measurements (see Sec. 2.2).

- Lab: The velocity in the direction of the flow was measured by a hot-wire anemometer. The data set consists of $8.48\mathrm{x}10^6$ points with a sampling frequency of $8\,\mathrm{kHz}$. The measured integral length scale is reported as $0.067\,\mathrm{m}$ (Fuchs et al., 2022). Details of the experiment are found in Renner et al. (2001).

## Appendix F:  CTRW model for the generation of wind fields

More detailed descriptions of the model are provided by Kleinhans (2008); Yassin et al. (2023); Mücke et al. (2011); Schwarz et al. (2019). Time series of the wind speed $u_i^{(\kappa)}(t)$ at each point $i$ of a defined grid are based on two coupled Ornstein-Uhlenbeck (OU) stochastic processes $u_r^\kappa(s)$ and $u_i^\kappa(s)$. Both processes are first generated in an intrinsic scale $s$. The super

index $\kappa$ accounts for the three directions of the wind $\kappa = [x, y, z]$. In our case, we generate wind speed time series only in the longitudinal direction $u^{(x)}$, so that $\kappa = (x)$. The two processes are defined as,

$$\frac{\mathrm{d}u_r^{(\kappa)}(s)}{\mathrm{d}s} = -\gamma_r(u_r^{(\kappa)}(s) - u_0^{(\kappa)}) + \sqrt{D_r}\Gamma_r^{(\kappa)}(s) \tag{F1}$$

and,

$$\frac{\mathrm{d}u_i^{(\kappa)}(s)}{\mathrm{d}s} = -\gamma(u_i^{(\kappa)}(s) - u_r^{(\kappa)}(s)) + \sqrt{D_i^{(\kappa)}}\Gamma^{(\kappa)}(s) \tag{F2}$$

where $\gamma$ and $\gamma_r$ are damping constants, $D$ and $D_r$ are diffusion constants; and $\Gamma(s)$ and $\Gamma_r(s)$ are Gaussian-distributed white noise. Next, the resulting Gaussian velocity signals $u_i^{(\kappa)}(s)$ are mapped to the physical time scale $t$ by means of an additional stochastic process as,

$$\frac{\mathrm{d}t(s)}{\mathrm{d}s} = \tau_{\tilde{c}, \alpha_L}(s). \tag{F3}$$

where $\tau_{\tilde{c}, \alpha_L}(s)$ is a Lévy-distributed process with characteristic exponent $\alpha_L$ and a cutoff value $\tilde{c}$. In the case of $\alpha_L = 1$, the intrinsic scales $s$ is equivalent to the physical time $t$ so that $u_i^{(\kappa)}(s) = u_i^{(\kappa)}(t)$. The time mapping process described in Eq. (F3) allows the key feature of the model which accounts for the intermittent behaviour of the wind speed time series. The intermittency is introduced by the Lévy-distributed sizes of the waiting times for the transformation from $s$ to $t$.

In Sec. 4 we investigated two CTRW data sets: CTRW-G and CTRW-NG. For the CTRW-G time series shown in Fig. 8 and Fig. 9(a) the Lévy exponent $\alpha_L$ is equal to 1 so that the waiting times of the intrinsic scale $s$ are constant and the statistics of $u(t)$ are Gaussian. For the CTRW-NG time series, we assumed $\alpha_L = 0.9$. By doing so, we introduce non-Gaussian features on the probability distributions. Further values of the parameters for generating the fields are given in Appendix E.

## Appendix G:  Spatial coherence of $T_c$

The spatial coherence of the CWS periods has been preliminary investigated. Fig. G1 shows the results of evaluating the simultaneity of events $T_c > T_{min}$, occurring at different heights of the FINO data, and conditioned on a reference height $\tilde{H}$. Exemplary, Fig. G1 shows the case when considering the reference height $\tilde{H} = 90\,\mathrm{m}$, and $T_{min} = 30\,\mathrm{s}$. Then, for each event $T_c > 30\,\mathrm{s}$ at $90\,\mathrm{m}$, the occurrence of simultaneous events $T_c$ at the remaining heights $H$ is evaluated. A black line is drawn when an event $T_c$ is measured at the corresponding $H$.

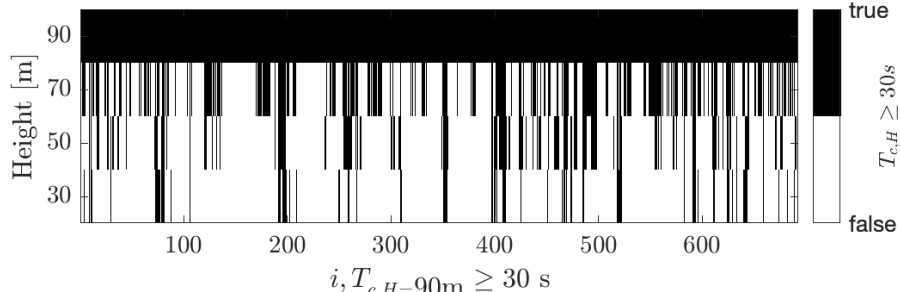

**Figure G1.** Events $T_c > T_{min}$ at different heights, conditioned on $\tilde{H} = 90$ m. First, the reference height $\tilde{H}$ is defined. Next, for each $i$ event $T_{c,i} > T_{min}$ at $H = \tilde{H}$, the occurrence of $T_c$ at the remaining heights $H = [70, 50, 30]$ m is evaluated. Black lines depict the occurrence of an event. The $T_c$ at all heights $H$ are conditioned so that $T_c > T_{min}$. For the example in this figure, $T_{min} = 30$ s and $\tilde{H} = 90$ m.

The results show that most of the events are not coherent over the four heights $H$ and confirm the appearance of localized structures. In fact, for the example shown, $37\%$ of the events at $H = 90$ m are happening simultaneously at $H = 70$ m. This number decreases to $11\%$ when comparing the CWS periods between $H = 90$ m and $H = 30$ m. The same evaluation for coherent events has been performed for different values of $T_{min}$ and reference heights $\tilde{H}$.

*Author contributions.* Daniela Moreno: Development of the code for measuring the periods of constant wind speed from different data sets; generation of the synthetic wind data, analysis of the data and writing the core of the document. Jan Friedrich and Matthias Wächter: Review, analysis, discussion of the results, and contributions to the text. Jörg Schwarte: Discussion on the results from the manufacturer/operator perspective. Joachim Peinke: Extensive understanding of the method, analysis of the results, supervision, reviewing, and editing the text.

*Competing interests.* The authors declare no conflict of interest. An author is a member of the editorial board of journal WES.

*Acknowledgements.* We gratefully appreciate the valuable discussions with our partners, the Institute for Mechanical and Industrial Engineering Chemnitz and Nordex Energy SE, involved in PASTA project (Precise design methods of complex coupled vibration systems of modern wind turbines in turbulent conditions). The aim of this work was initiated as a hypothesis for the challenges discussed within the project.

This work has received funding from the German Federal Ministry for Economic Affairs and Climate Action according to a resolution by the German Federal Parliament (PASTA, 03EE2024). Jan Friedrich acknowledges funding by the European Union within the project Atmospheric FLow, Loads and pOwer for Wind energy (FLOW, HORIZON-CL5-2021-D3-03-04, grant no. 101084205).

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
