# Peer review of "Periods of constant wind speed: How long do they last in the turbulent atmospheric boundary layer?"

_Wind Energy Science, 2024_

## Referee Comment (RC2)

[referee-annotated manuscript omitted]

---

## Author Comment (AC1)

**Response to Referee 1**

**Periods of constant wind speed: How long do they last in the atmospheric boundary layer?**

Referee's comment (RC) in blue
Author's comment (AC) in black

*In gray-italic: text from the revised version of the manuscript.*

AUTHORS:

Dear Referee, thank you for highlighting the importance of our research. We appreciate your feedback. In the following, we would like to address the open questions and comments you have posted.

We use the following abbreviations: Constant wind speed (CWS), Period of constant wind speed ($T_c$), Atmospheric Boundary Layer (ABL), Wind turbine (WT), Probability Density Functions (PDFs).

**GENERAL COMMENTS**

REFEREE:
The paper is strong in the technical aspects with well selected analyses to possibly support the hypothesis that the distribution of the CWS event durations can be described with an exponent.
 While it is an interesting topic to address, and important to highlight that the standard turbulence model is inadequate, the paper lacks proper motivation. More information would be needed to determine how these events may increase the WT loads. Are they coherent, i.e. do they occur throughout the heights enveloping the rotor area? Which design case of the IEC WT design standard do they fit into, or is a new design case needed (Introduction, L31)? How can the knowledge of these events help the WT and wind plant controller? Can an event be predicted from past few seconds of data?
 A major shortcoming is that the period of the observation used for the analysis is too short. Much more data must be analyzed for a meaningful publication. One could, for example, object against using the statement "conclusive", not once but twice: in the Abstract, and in the section 4.2, page 11. When the period of data collection is extended, the data could then also be separated by wind direction, surface heat flux, and possibly expose additional properties.
 The curvature of the spectrum (figures 5, 6, 7) indicates imperfect power law. There is curvature present even at the long durations, which does not help the results being conclusive. One would need to propose a theory at least trying to explain the power-law with physical characteristics of the boundary layer (stability, surface roughness, ...) and then blame the disagreement on incomplete

AUTHORS:

Based on your comment that the motivation of our research was insufficiently stated, we have revised, rearranged and incorporated additional statements in the Introduction (Sec. 1) to reinforce the significance of our study. In the new version of the manuscript, we contextualize in a more clear way, compared to the original version, the periods of CWS within the general characterization of turbulence. We also explicitly explain the potential relevance of periods of CWS for WT loads.

Starting in L.35, we motivate our research within the framework of general characteristics of turbulent flows.

From L.60 to L.69 we better formulate our hypothesis regarding the possible increased loads on a WT induced by a period of CWS with certain characteristics. This can be the case when a localized period of CWS occurs on a limited area on the rotor plane.

You have commented on the coherence as a relevant feature of the CWS periods for assessing their potential impact on WTs. Indeed, we have investigated coherent CWS events through the analysis of the FINO1 wind measurements at different heights (see details of the data in Sec. 2.2 of the manuscript). The results of our preliminary investigations suggest that the periods of CWS appear localized at different heights. An example of the analyses is shown in Fig. 1 in this document. Nonetheless, further studies from meteorological mast arrays (e.g., GROWIAN data or the WiValdi test site) should shed more light on the spatial coherence of these events.

[Figure]

Figure 1: Events $T_c > T_{min}$ at different heights, conditioned on $\tilde{H} = 90$m. First, the conditioning height $\tilde{H}$ is defined. Next, for each $i$ event $T_{c,i} > T_{min}$ at $H = \tilde{H}$, the occurrence of $T_c$ at the remaining heights $H=[70, 50, 30]$m is evaluated. Black lines depict the occurrence of an event. Note that $T_c$ at all heights are conditioned so that $T_c > T_{min}$. For the example in this figure, $T_{min} = 30$s and $\tilde{H} = 90$m.
In this case, 37% of the events at 90m are happening simultaneously at 70m. This number decreases to 11% when comparing the periods of CWS between 90m and 30m. The same evaluation for coherent events has been performed conditioned by different values of $T_{min}$ and reference heights $\tilde{H}$.

We have referred to our preliminary investigation on the coherence of periods

of CWS in L.299 in the Outlook of the manuscript:

*Accordingly, preliminary investigations (detailed in Appendix G) suggest that the periods of CWS show a tendency to be localized at different measurement heights, and therefore, may become of particular interest for turbines with larger diameters.*

Additionally, we now have provided details about the coherence investigation in the Appendix G: "Spatial coherence of $T_c$" of the manuscript, including Fig. 1 shown above.

Concerning the issue of the insufficient length of the investigated period, we agree that the period of 4.6 days for the analysis in the original manuscript might have been too short. Therefore, we re-analyzed for a longer period (roughly one year; see the revised version of the manuscript). Our major finding, i.e., the power-law behavior of the tails of the probability density function (PDF) of the CWS periods, is confirmed with even more accuracy. The apparent curvature in the original log-log-plot of the PDFs decreased as well, which led to more accurate power law fits (see Figs. 3 and 4 in the manuscript).

Regarding a potential theoretical explanation for curvature effects in the power laws, we must emphasize that our findings here are entirely empirical, and future work has to be devoted to describing such periods of CWS in the ABL within a coherent statistical description.

L.301 in the Outlook refers to the need of a complete description of the CWS periods:

*Future work has to be devoted to assessing the relevance of the empirically observed power-law behavior of periods of CWS on turbine loading. For that, the complete statistical parametrization of periods of CWS, in both time and spatial domains, should be assessed and improved(...).*

You have posed an interesting question about a potential IEC Design Load Case (DLC). To our knowledge, there is currently no DLC that addresses non-coherent spatial conditions during the operation of the turbine. We agree that considering a standardized framework for the load assessment of the loads under such turbulent conditions would be highly relevant for the turbine design and certification processes. However, before adding such a feature to the IEC design guidelines, intensive load analyses with a specific WT model and control strategies have to be performed together with industrial partners.

**SPECIFIC COMMENTS**

**Abstract**
L3: please elaborate on "particular dynanmic responses"

With "particular dynamic responses" we refer to unexpected responses (e.g. loads, deflections, resonances) induced by periods of CWS with specific characteristics. An explanation of our hypothesis is given in L.63-L.65:

*A more entangled case might occur when resonant or near-resonant dynamics appear for specific periods of CWS, over which the resonance can be strongly excited. In particular for the larger WTs, the CWS periods may be restricted to a sub-area of the rotor plane. In this case, resonant dynamics exhibiting 3P oscillations may be amplified.*

L6: what is meant by "the challenging power law behaviour" and why is this introduced with a reference to extreme events? Extreme events are not mentioned anywhere in the paper, other than extremely long CWS duration. Speaking of extreme events ... please verify if they perhaps follow any of the typical extreme event distributions

Thank you for this important comment. We admit that the term "challenging power law behavior" might have been misleading. We intended to discuss the divergence of moments of certain orders dependent on the power law's exponent in the Pareto distribution. The Pareto distribution belongs to the class of heavy-tailed distributions. Regarding the reference to extreme events, we refer to extreme events in the sense of very long periods of CWS, which is the focus of our investigation.

We would also like to thank you for your suggestion regarding considering typical extreme-value PDFs. However, in our study, we investigate the statistics of all measured periods $T_c$ and analyze the resulting tails of their PDFs. By doing so, we clearly observe extreme events but do not follow explicitly the procedure of extreme value statistics (i.e., for $T_c$ larger than a threshold). In that way, extreme value distributions can not be applied.

As our new results based on a much longer data set suggest power law exponents rather far from the mentioned criticality, we no longer consider this aspect significant for the manuscript. Therefore, we removed L.6 in the original version of the manuscript.

We included L.55-L.59 for describing the peculiarity of the power law behavior:

*A characteristic feature of a power law distribution is the absence of an intrinsic scale, i.e., the probability of observing a realization larger than $\xi T$ is $\xi^{-\alpha+1}$ times the probability of observing a realization larger than $T$; independently of the value of $T$. The far-tail regime of many distributions occurring in complex systems is assumed to exhibit power-law behavior [Laherrere and Sornette, 1998]. In the context of wind energy, for instance, a Pareto distribution has been tested to extrapolate the response of a multi-megawatt wind turbine generator [Dimitrov, 2016].*

**Section 2.1**

L109: It is hard to imagine how would a WT know that a CWS event is imminent and switch into the appropriate control mode in practice.

Thank you for your question. We admit we do not have a concrete answer to how the control strategy would be in an imminent period of CWS. In L.126 (L.109 in the previous version) we refer to the control practices of the WT for introducing a reference value for defining the factor $A$ in $\varepsilon = A \cdot \sigma_u$ for measuring the periods of CWS within the characterization of the turbulent wind. Our intention is not to comment on or propose a reference parameter for identifying and reacting to a CWS structure in the context of WT control protocols. However, as a side remark, we believe that forecasting periods of CWS is not strictly necessary. Instead, the control system should be designed to react to long-lasting undamping events, which might be the effect of a period of CWS on the WT.

**Section 3.4**

L158: Good that the effect of the threshold amplitude is analyzed. Would it not be appropriate to add the resulting alpha exponent as another row in Table 3 and so enable easy comparison of different alphas

The results of $\alpha$ have been included in Table 3. Thank you for the suggestion.

**A new version of the manuscript is provided along with a diff file**.

**References**

N. Dimitrov. Comparative analysis of methods for modelling the short-term probability distribution of extreme wind turbine loads. *Wind Energy*, 19(4): 717–737, 2016.

J. Laherrere and D. Sornette. Stretched exponential distributions in nature and economy:"fat tails" with characteristic scales. *The European Physical Journal B-Condensed Matter and Complex Systems*, 2:525–539, 1998.

---

## Author Comment (AC3)

**Response to Referee 2**
**Periods of constant wind speed: How long do they last in the atmospheric boundary layer?**

Referee's comment (RC) in blue
Author's comment (AC) in black

*In gray-italic: text from the revised version of the manuscript.*

Authors:
Dear Referee, thank you for your comments and recommendations. In the following, we would like to answer the points you have addressed.

We use the following abbreviations: Constant wind speed (CWS), Period of constant wind speed ($T_c$), Atmospheric Boundary Layer (ABL), Wind turbine (WT), Probability Density Functions (PDFs).

**GENERAL COMMENTS**

This work addresses an academically interesting subject, but in its current form the draft unfortunately has some serious deficiencies. This includes speculation about extreme winds and turbine loads; the manuscript does not connect (directly) to extremes nor give a solid basis for loads. More importantly perhaps, its analysis of only 4.6 days of measurements cannot be used to justify statistics for loads accruing over the multi-decadal lifetime of a turbine, especially for extremes (see e.g. Dimitrov et al., 2018). The large variations in "critical" exponent for calm-durations may be a affected by this, but it is not clear. The lack of citation (or even use) of previous mathematical developments for persistence statistics is also an issue, especially given the conclusions about $T_c(\varepsilon)$; e.g., Hurst exponents (or even fractal dimension) for such have been explored by numerous authors. The fitting of a power-law over ranges where log-log plots show significant curvature, as well as the subsequent neglect of both the range of application or functionally different form of PDF – and implied lack of convergence for the power-law given the resultant exponents – are serious issues to be rectified, requiring more analysis. Overall the work on systematically quantifying the duration of 'persistent' periods offshore (and possibly capturing their statistics via a CTRW or other model) is interesting, and could merit publication; but to connect this with extremes would likely require significantly more work, which would be the subject of another separate publication(s).

We agree with the referee that the analysis period of roughly five days for the constant wind speed (CWS) events might have been too short. Therefore, in the revised version of the manuscript, we chose a more extended analysis period of roughly a year. This extension clearly supports our main findings,

i.e., the existence of power-law behavior for the probability density function of the periods of CWS.

Furthermore, we must stress that "extremes" in the manuscript refer to comparatively long periods of quiescent wind speeds, i.e., the opposite of extreme wind field fluctuations. Nonetheless, we agree that the relation between such periods of CWS and their potential effect on turbine loading could be more precise. Therefore, starting on L.60 in the Introduction, we formulate in a more comprehensible way, our hypothesis regarding the possible increased loads on a WT induced by a period of CWS with certain characteristics. In particular, a relation to turbulent-non turbulent transitions [Neuhaus et al., 2024, Lobo et al., 2023] within the operation of the WT is mentioned. This can be the case when a localized period of CWS occurs on a limited area on the rotor plane. In Appendix G we provide evidence of such localized events. Starting in L.297 in the Conclusions, we discuss again the potential effect of periods of CWS on particularly large WTs.

We want to emphasize that we appreciate the referee's comments on the persistence events (excursions and zero-crossings). We include several references to these in our manuscript (see Introduction). However, even though persistent events such as inter-arrival times of excursions and/or zero-crossings, and our definition of periods of CWS exhibit certain similarities, they are not the same (please refer to Fig. 1 in this document and our response to your comment on l.1 in the SPECIFIC COMMENTS section).

[Figure]

Figure 1: Illustration of excursions, zero crossings, and periods of CWS ($T_c$). In the top-plot, the exemplary wind speed time series are normalized to zero mean and standard deviation 1. The blue area defines the thresholds $\pm\mathcal{U}$ for considering the excursions, represented by blue crosses. $\mathcal{U}$ is defined as $2\varepsilon$. The red crosses depict the zero crossings. The grey rectangles show the periods $T_c \geq 10s$ measured with $\varepsilon = 0.3$. The blue and red lines in the bottom plot depict the resulting inter-arrival times for the excursion measured at $+\mathcal{U}$, and the zero-crossings, respectively. For comparison, the periods $T_c > 10s$ are replotted as black lines. The inter-arrival times for the excursions and zero-crossings in the plot are filtered to be longer than 10s, as are the periods $T_c$.

To our knowledge, the analysis of periods of CWS in atmospheric turbulence and the obtained power law behavior of their PDFs has not been investigated before.

Furthermore, based on your suggestion of analyzing a more extended period of roughly a year of FINO data, the issue of potential curvature in the log-log plots of the PDFs shown in the original manuscript could be resolved.

Our reasoning for including a discussion of a wind field model (in this case, the CTRW model) in this context of periods of CWS is the following: Fig. 7 of the revised manuscript shows that the IEC Kaimal wind field model significantly underestimates the occurrence of long periods of CWS from atmospheric wind. For a future assessment of the relevance of periods of CWS for wind turbine loads, however, a wind field model that could reproduce the empirically observed power law behavior is indispensable. Therefore, we believe that Sec. 4.2 in the manuscript complements our findings in Sec. 4.1 quite well.

**SPECIFIC COMMENTS**

l.1: this is not a "non-investigated" topic; see e.g. Majumdar's "Persistence in nonequilibrium systems" (1999), or in the ABL, e.g. Chowdhuri, et al. (2020), https://doi.org/10.1063/5.0013911.
Thank you for recommending the literature. Persistence events including inter-arrival times between excursions and zero-crossing analyses have been investigated and reported in the literature. In fact, in the Introduction of the original version of the manuscript we have already referred to the zero-crossings analysis. Now we also refer to the concept of excursions.

However, even though their definition may be similar with respect to the periods of CWS, they are not directly related. Please see Fig. 1. In the top-plot, the zero-crossings of the zero-mean and normalized time series of $(u(t) - \bar{u})/\sigma_u$ are shown by red crosses. In [Chowdhuri et al., 2020], persistence is defined as the inter-arrival times between the zero-crossings. Comparatively, the grey rectangles mark the measured periods of CWS ($T_c > 10$s). For a clearer comparison, the bottom-plot shows the duration of the inter-arrival times between the zero-crossings (red lines). As observed, persistence events and periods of CWS are not the same. A period of CWS might enclose several zero-crossings, as well as several periods of CWS might be embedded inside an individual inter-arrival between zero-crossings.

Starting in L.35 in the Introduction, we now rigorously provide a context for the periods of CWS within the general characterization of turbulence. We introduce related concepts such as persistence, extreme winds and zero-crossings.

We emphasize the difference between persistence events (excursions and zero crossings), and periods of CWS in the Introduction L.49-L.52 of the manuscript:

*It is worth noting that even though the inter-arrival times of both, excursions and zero-crossings, refer to structures between particular turbulent states, they do not correspond to the periods of reduced turbulent amplitudes, in which we are interested. Further details of the differences between CWS periods and inter-arrival*

*times between excursions and zero-crossings are shown in Appendix A*

We have now included the Appendix A: "Periods $T_c$ vs persistence events" in the manuscript. There we incorporate Fig. 1 shown above in this document.

We have included the reference to [Chowdhuri et al., 2020] in the Introduction.

l.3: jets are not shown here to be "characteristic wind field structures" responsible for or related to constant wind speed periods, nor has this been referenced (shown elsewhere).

Here, we have used the term "jets" as an illustration for the reader rather than as a specifically defined phenomenon.

l.5: how have the constant-wind speed periods been related to extreme events? This should be removed unless such connection has been shown.

Please refer to our response to your GENERAL COMMENTS. In the second paragraph we stress that by "extreme events", we mean very long periods of CWS.

l.6-7: extreme events are known to follow "fat-tail" or power-law behavior, this is not new; "show" should be "confirm".

Thank you for this suggestion. We changed "show" to "confirm" in L.6 in the abstract.

l.45: wasn't the event-measurement approach developed in 2022, or how is this different?

It is true that a precursor of the approach was introduced in [Moreno et al., 2022]. We have actually removed this sentence from the manuscript. Our preliminary work ([Moreno et al., 2022]) is now disclosed in L.80:

*In a preliminary investigation [Moreno et al., 2022], the method for the assessment of such events from wind speed time series was presented and first results on the characterization of the periods of CWS in terms of their duration and probability distributions were also reported.*

Additionally, in L.122 we mention the difference, compared to [Moreno et al., 2022], of the method for measuring the events $T_c$ from time series of the wind $u(t)$.

*To introduce a systematic approach, in this paper $\varepsilon$ is defined to be proportional to the standard deviation of the wind speed $\sigma_u$. Then, $\varepsilon$ for fixing (...)*

l.65: "Such strong periods are expected to have a stronger influence on a WT" does not seem correct, given that the periods with weakest fluctuations will have less effect on wind turbine loads.

Yes, we agree this is a misleading statement. We have removed it from the manuscript. Thank you for the comment.

l.81-82: note that this (and Fig.1) is essentially the (reverse of) the generalized gust description of L.Kristensen et al (1991) following from Rice (1945).

Thank you again for calling attention to these important studies regarding extreme wind speeds and the waiting times between them. However, as stated in our response to your GENERAL COMMENTS, we do not address extreme winds but extreme (very large) periods of CWS. Even though the inter-arrival times between extreme winds (or gusts) seem similar to the periods of CWS, they do not refer to the same wind structure. Please see again to Fig. 1 in this document.

In the top-plot of Fig. 1, the blue area defines the thresholds $\pm\mathcal{U}$ for considering the extreme winds (also called gusts, or excursions). The blue crosses depict the times when such thresholds are crossed. Again, periods of CWS ($T_c$) larger than 10s are marked by the grey rectangles. For comparison, the threshold $\mathcal{U}$ for the excursions is defined as twice the threshold $\varepsilon$. In that way, a period $T_c$ inside $\pm\varepsilon$ would contain the size of the fluctuations defined inside the ranges $(0,\mathcal{U})$ or $(-\mathcal{U},0)$. In a similar way as depicted by the red lines for the inter-arrival times between zero-crossings (see response to your SPECIFIC COMMENT on l.1), the lengths of the inter-arrival times between excursions are shown by blue lines in the bottom-plot. In this case, the inter-arrival times are considered only for the positive threshold $\mathcal{U}$. As observed, the periods of CWS are not equivalent to the inter-arrrival times between excursions. A period of CWS might enclose several excursions, as well as several periods of CWS might be embedded inside an interval between two consecutive excursions.

Refer to the modifications to the manuscript mentioned in our response to your SPECIFIC COMMENT on l.1.

We have included the reference to [Rice, 1944] and [Kristensen et al., 1991] in the Introduction of the manuscript.

l.132 (also eq.2, l.107-109): which averaging interval is used for the $\sigma_u$ here? Is it for each 10-minute period, or over the entire 5-day dataset?

We now make it clear in the text, L.129:

*In Eq.(2) and through this document, we refer to $\bar{u}$ and $\sigma_u$ as the values calculated over 10-minute periods unless a distinction is clearly stated*

l.133: do you mean that taking A=0.3 means here that $I_u \leq 0.02$ due to the mean wind speed value (presumably $\approx$15 m/s)? This is not clear. (Or why not pick A based on Iu?)

Since now we are analyzing a longer data set from the FINO data with variable values of $\bar{u}$ and $\sigma_u$, this sentence is not valid anymore. Therefore it has been removed from the manuscript.

l.134: The values of $\varepsilon$ are not provided in Table 2.

This sentence has been removed from the manuscript. As the value of $\varepsilon$ is now calculated for each individual 10-min period (based on $\sigma_u$), there is not such a single value of $\varepsilon$ at each height $H$.

l.161: There is not a simple 'clear power-law decay' for all values of A; in Fig.5 one sees a curved line which becomes straighter for rarer longer $T_c$.

The apparent curvature in the original log-log-plot of the PDFs decreased with the larger data set. More accurate power law fits are obtained (see Figs. 3 and 4 in the manuscript).

l.162: the finding $2 < \alpha < 3$ also means that the PDF can't integrate to 1 (its normalization constant is undefined), consistent with the fact that the lines are curved particularly at more common (shorter) $T_c$. The distribution is more like a stretched exponential or some other extreme value PDF. You cite Clauset/Shalizi/Newmann2009, but are still fitting a straight line on log-log (which they advised against) despite evidence that the power-law has a limited range of application for $T_c$.

Thank you for this critical comment. We follow the recipe for analyzing power-law distributed data provided in [Clauset et al., 2009], including estimating a lower bound on power-law behavior.

We now make it clear in the text, L.165:

*The corresponding exponents $\alpha$ are calculated following the procedure proposed by [Clauset et al., 2009] and described in Appendix D.*

We would also like to thank you for mentioning extreme-value PDFs. However, in our study, we investigate the statistics of all measured periods $T_c$ and analyze the resulting tails of their PDFs. By doing so, we clearly observe extreme events but do not follow explicitly the procedure of extreme value statistics (i.e., for $T_c$ larger than a threshold). In that way, extreme value distributions can not be applied.

Fig.5: the horizontal axis has 1/3 empty space, and should be reduced; one can also then see more clearly where the curves become flat or not.

Thank you for noticing. We modified Fig. 5 (Now Fig.4) in the revised version of the manuscript.

Fig.9: there is no scale on the $p(T_c)$ axis, spanning (apparently) 8 orders of magnitude; please include scaling factors and reduce empty space (extra 3-4 orders of magnitude on vertical axis and 50% horizontally).

Thank you for noticing. We added a scale to the axis in Fig. 9 and reduced the empty space.

Fig.9b caption: mention dotted red horizontal line here also.

Thank you for the recommendation. The line is now mentioned in the caption.

l.49: the mean and variance constraints demand that $\alpha < 2$ and $\alpha < 3$ respectively, but these are weaker constraints than that on the PDF itself ($\alpha < 1$); thus I recommend removing this phrase.

Thank you, we removed this phrase.

l.245: "very long" needs to be quantified, especially because you have used only a few days of data.

Thank you for the comment. The values of the longest periods of CWS are provided in Table 2 and Table 3 in the manuscript. Additionally, the comparison shown in Fig. 7 in the manuscript allows the relation of the lengths of the periods $T_c$ to the large eddy turnover time ($T_{int}$) of the flow.

In the conclusion, L.269,

*It is shown that the probability distributions $p(T_c)$ for offshore data exhibit a power law decay $p(T_c) \propto T_c^{-\alpha}$ for very long events (i.e. hundreds of seconds).*

and L.276,

*We found examples of $T_c$ significantly larger than 100s, which correspond to spatially extended structures over sizes larger than 1km (...)*

refer to the actual lengths of the periods.

l.246-7: you state "offshore conditions maintain a more unperturbed ABL", but compared to what? What (normalized) metric are you invoking to state this?

We mean compared to onshore conditions. see L.132:

*We expect offshore wind to provide a better representation of undisturbed, or less disturbed conditions within the ABL compared to onshore data. Therefore, the possible effects of onshore orographic conditions on the CWS structures are diminished.*

L.272 has been modified to accordingly:

*Given that offshore conditions maintain a more unperturbed ABL compared to onshore, we demonstrated (...)*

l.255: "proved the turbulent nature of the wind speed" does not make sense as a statement alone, and without saying how. What does this mean, is this statement necessary?

By proving the decay $E(f) \propto f^{-5/3}$ in Sec. 3.4 we confirm the turbulent nature of $u(t)$ during the periods $T_c$.

L.280 has been modified:

*Based on the spectral properties, we proved the turbulent nature of the wind speed $u(t)$ during the periods $T_c$.*

l.261-2: the spectral gap is an effect, it does not cause effects; maybe state "phenomena related to the spectral gap".

Thank you for the correction. It has been included in the text. L.286

*(...) or whether they are indeed consequences of larger-scale interactions of the atmospheric boundary layer, like phenomena related to the spectral gap [Larsén et al., 2016].*

l.268-270: how do weak-turbulence periods cause "critical loads"? I suggest removing this phrase, unless you can explain and justify. Similar for "resonance". The justification on l.281 and later is ok (though it is a weaker effect than anti-correlated coherent turbulence across the rotor).

You are right. There is no justification for this rigorous statement. We have removed this sentence from the manuscript.

l.298, eq.(A1): there needs to be some minimum x and/or alpha<1, otherwise C is undefined, i.e., the integral of p(x) from 0 to infinity does not converge. You state simply "alpha>1" later in Appendix B, but this constraint should be mentioned here. In App.B you do assume an $x_{min}$, without explanation.

You are right. A more complete definition of the variables in Eq. (A1) has been now introduced.

L.329 to L.332 have been modified:

*... for $x \geq x_{min}$ with the characteristic exponent $\alpha$ and a constant $C = e^c$. The minimum value $x_{min}$ holds for the lowest limit of the power-law. The exponent $\alpha > 1$, otherwise $\int_0^\infty x^k p(x)$ does not converge.*

l.319: $\leq$ should be $\geq$ here; i.e., the mean and variance constraints demand that alpha<2 and alpha< 3 respectively, but these are weaker constraints than that

on the PDF itself (alpha<1).Given the latter, I would suggest removing the statement about mean and variance, unless you wish to move it to App.B add conditions on it with the use of $x_{min}$.

Thank you for the recommendation. In fact it made more sense to move the whole paragraph about the statistical moments to Appendix C (Appendix B in the previous version of the manuscript). That of course includes the sentences about the mean and the variance of $x$.

l.322-3 is the sentence about seismic events relevant here?

Not really, the sentence has been removed. Thank you for the recommendation.

l.370: should there be a comma between 0.9 and 1 for $\alpha_L$?

Thank you for noticing. It has been corrected.

**A new version of the manuscript is provided along with a diff file**.

**References**

S. Chowdhuri, T. Kalmár-Nagy, and T. Banerjee. Persistence analysis of velocity and temperature fluctuations in convective surface layer turbulence. *Physics of Fluids*, 32(7):076601, 07 2020. ISSN 1070-6631. doi: 10.1063/5.0013911.

A. Clauset, C. R. Shalizi, and M. E. J. Newman. Power-Law Distributions in Empirical Data. *SIAM Review*, 51(4):661–703, 2009. ISSN 00361445, 10957200.

L. Kristensen, M. Casanova, Courtney M.S., and Troen I. In search of a gust definition. *Boundary-Layer Meteorology*, 55(1-2):91–107, 1991. ISSN 0006-8314. doi: 10.1007/BF00119328.

Xiaoli G. Larsén, Søren E. Larsen, and Erik L. Petersen. Full-scale spectrum of boundary-layer winds. *Boundary-Layer Meteorology*, 159:349—371, 2016. doi: 10.1007/s10546-016-0129-x.

B. A. Lobo, Ö. S. Özçakmak, H. A. Madsen, A. P. Schaffarczyk, M. Breuer, and N. N. Sørensen. On the laminar–turbulent transition mechanism on megawatt wind turbine blades operating in atmospheric flow. *Wind Energy Science*, 8 (3):303–326, 2023. doi: 10.5194/wes-8-303-2023.

D. Moreno, J. Friedrich, M. Wächter, J. Peinke, and J. Schwarte. How long can constant wind speed periods last in the turbulent atmospheric boundary layer? *Journal of Physics: Conference Series*, 2265:022036, 05 2022. doi: 10.1088/1742-6596/2265/2/022036.

L. Neuhaus, M. Wächter, and J. Peinke. The fractal turbulent–non-turbulent interface in the atmosphere. *Wind Energy Science*, 9(2):439–452, 2024. doi: 10.5194/wes-9-439-2024.

S. O. Rice. Mathematical analysis of random noise. *The Bell System Technical Journal*, 23(3):282–332, 1944. doi: 10.1002/j.1538-7305.1944.tb00874.x.

---

## Author Response (AR2)

**Response to Editor**

**Periods of constant wind speed: How long do they last in the atmospheric boundary layer?**

Editor's comment (RC) in blue
Author's comment (AC) in black

*In gray-italic: text from the revised version of the manuscript.*

AUTHORS:
Dear Editor, we appreciate your comments and recommendations. In the following, we would like to address the open questions you have posted.

We use the following abbreviations: Constant wind speed (CWS), Atmospheric Boundary Layer (ABL), Wind turbine (WT), Probability Density Functions (PDFs). In this version of the manuscript we have introduced a new abbreviation for the Constant Speed Range (CSR).]

**GENERAL COMMENTS**

EDITOR:
1. You kind of argue that the CWS periods cannot be reproduced by standard turbulence models; if their pdfs/statistics are quite local, the question is whether they can be reproduced by weather models that one can run routenly anytime anywhere.
   Whether a weather model can reproduce very long CWS periods is a very interesting question. The statistical characterization of data from such atmospheric models would provide insight into their potential use for the analysis of the effect of CWS structures on WTs. However, this analysis is beyond the scope of the current investigation. Valuable future work on the topic could be based on weather-modelled data (e.g., ECMVF, WRF).

Lines 309 - 311 have been added to the manuscript as part of the outlook:

*Another interesting aspect for future work would be the statistical analysis of CWS periods from weather-modelled data (e.g. ECMVF, WRF models). The results would reveal whether such larger-scale models can reproduce the CWS structures within the atmospheric forecasting.*

2. I think you overstress the word "conclusive" and "conclusions" throughout the manuscript without the need for it. Such use does sound like you want to settle the discussion but there is no need for it. There is a section "Conclusions" where you should focus your conclusions and concluding remarks. Please find such statements and rephrase.

We very much appreciate this comment. We agree that the use of the 'conclude' and 'conclusive' statements is excessive. We have carefully revised them throughout the paper. The changes can be identified in the diff file.

3. In Table 1 the standard deviation is taken as the average over all 10-min standard deviations, so it is not turbulence; if this info is not used I advice to remove table 1 and its details.

Yes, you are right. The Table was useful in the previous version of the manuscript. However, at this point it does not provide relevant information to the investigation and the results in the paper. The table has been removed from the manuscript.

4. In code and data availability you tell the readers that this could be done upon request. I recommend you already make these datasets and codes available in a repository or so. The "upon request" statement does not encourage people to use your findings and methods and we all have face no responses when trying to reach authors. Instead, inmidiate code/dataset publication will make your work open to the community (and you will be way much more cited).

Thanks a lot for the suggestion. We are working to make the codes/functions publicly available. Work in progress at:
https://github.com/danielamorenom26/cws.
We hope the codes will be available before the publication of the paper. In that case, the section *Code availability* would be modified.

**SPECIFIC COMMENTS**

You have general an issue with the references. They should be fully in brackets when they are passive, e.g., if one says: ".... compared to the standard models (IEC, 2019)." But the brackets should only cover the year when the reference is active, e.g., if one says "... as demonstrated by Moreno et al. (2019), the amount of energy...". Please revise all of your references.

Thank you for the constructive explanation. All the references have been accordingly revised in the manuscript.

Also in many instances you have an issue with the "math" mode, so please revise thoroughly. E.g., line 44 you have for H=90m; there should be a small space between H, the = sign, the 90 and the unit.

Thank you for highlighting the issue. All the "math" entries in the document have been revised.

In line 4 you use the word extreme that seems contradictory to a constant wind speed period.

The word "extreme" has instigated several discussions in the context of our investigation. As we analyze the statistics of the lengths of the CWS periods, an extreme event in this case corresponds to a very long CWS period.

In Appendix A of the manuscript, we recall the concept of wind speed excursions (i.e. wind speed $u(t)$ exceeding certain thresholds) which are often called extremes. Under such a definition of an extreme, we agree that a CWS period might be oppositely related. However, it is not rigorously defined.

In line 5 you say that the statistics of CWS are an intrinsic feature; a feature of the ABL could be the CWS periods but not their statistics.
Corrected in the manuscript.

Line 11 maybe replace "a typical multiscale effect. Given the conclusive results," by "multiscale."
Replaced in the manuscript.

Line 21 delete "extensively"
Deleted in the manuscript.

Line 59 to extrapolate the response of the MW turbine to what?
This sentence has been reinforced in the manuscript (still Line 59):

*In the context of wind energy, for instance, a Pareto distribution has been tested as an extrapolation method to estimate extreme loads on a multi-megawatt wind turbine generator [Dimitrov, 2016] with a 1-month return period.*

Lines 84-86: the sentence along these lines is at this point weird. One can use a standard spectral model to generate a time series characterized by specific turbulence parameters; so in principle, one could manage to synthetically produce a time series with small "turbulent amplitudes" (by playing with the spectral model parameters) so that the fluctuations do not surpass the threshold of choice.
Yes, we agree with your statement. In principle, the generation of time series with small amplitudes is possible from a spectral model. However, here the analysis of data from the IEC standard model aims to investigate the characteristics of the CWS periods within the wind fields currently used for wind turbine simulations. Modifying the characteristics of such standard wind fields is not our goal.

Eq. 2 the dot is normally a dot product which you do not imply here
The dot is removed from the equation and from other instances where it was wrongly used.

Table 2 summarizes some first results; what is the interpretation of these? Is this what one expects?
Thank you, a comment on the data in the table (Now Table 1) is quite appropriate. We included a sentence in the manuscript (Line 151.)

*From the values in Table 1, comparable $\overline{T_c} \approx 4\,s$ and $\sigma_{T_c} \approx 3\,s$ are obtained for the four heights $H$. More interestingly are the longest measured CWS periods $T_{c,max}$ at each height $H$. Periods with lengths up to $T_c \approx 40\,\sigma_{T_c}$ which correspond to more than $100\,s$ are measured.*

Line 180: in line with general comment 2 "We conclude that…" At this point you cannot conclude this; maybe you can hypothesize that these are the type of distributions.
Modified in Line 179 of the manuscript.

*This confirms our hypothesis on the Pareto-like distributions of $p(T_c)$ for large $T_c$ already observed in Fig. 3*

Lines 183-184 if you observe that by increasing A, alpha and the statistics of Tc change, then don't you know already you are not dealing with laminar flow?
Hopefully, the answer to the next comment will clarify this one. The change in the statistics of $T_c$ for different factors $A$ is not directly related to the turbulent nature of $u(t)$.

Lines 184-187 I am not sure that is really clear how the wind speed time series is selected for the spectral analysis. Do you only take a 5-day time series if there is at least one period of CWS with Tc>10 s, or do you look for all times where Tc>10 s and take the 5-day time series around it?
We modified the explanation of how the wind speed time series are selected for the spectral analysis. It should be clear that the analysis is done with excerpts of time series $u(t)$ within the CWS periods. Starting at Line 185:

*The spectra $E(f)$ are calculated from the extracted time series of $u(t)$ during CWS periods larger than $10\,s$. A time window of roughly five days was considered for extracting the definite time series $u(t)$ during $T_c > 10\,s$.*

Figure 5 why are not all the lines for each height start at the same frequency in the left part of the plot. And are not these 5-day periods, and so the lowest frequency should be much lower? Maybe by explaining better the previous comment this graph becomes much clearer
Hopefully, it is now clear what is shown in Figure 5. The intervals $u(t)$ (within CWS periods) have different lengths.

Figure 6: all frames can be combined in one single graph
The three frames have been included in a single plot. Thanks for the suggestion.

Line 230: in line with general comment 2 "We have shown conclusive…" Well you have done some good analysis so far for the observations but you only tried one standard wind model with one set of parameters (see my previous comment of lines 84-86)
Modified in Line 231 the manuscript.

*We have shown results on the distributions of CWS periods $p(T_c)$ in the ABL and their underestimation by the IEC standard Kaimal wind model...*

**A new version of the manuscript is provided along with a diff file**.

**References**

N. Dimitrov. Comparative analysis of methods for modelling the short-term probability distribution of extreme wind turbine loads. *Wind Energy*, 19(4): 717–737, 2016.

---

## Author Response (AR3)

**Response to Referee 2**

**Periods of constant wind speed: How long do they last in the atmospheric boundary layer?**

Referee's comment (RC) in blue
Author's comment (AC) in black

**GENERAL COMMENTS**

The authors have made responses and changes addressing most reviewer comments, and the manuscript has improved considerably. The longer measurement period also really helps the analysis and justification. The statistics of normalized Tc, and subsequent power-law exponents, now look quite reasonable (new Fig.3) and potentially useful.

There are some details missing around the scales used with synthesized turbulence from the Kaimal model, as well as the integral scales assumed. There also appear to be inconsistencies between the integral time and length scales (see details below). For consistency and replicability, these need to be addressed.

There are new/remaining language issues within the updated draft, which should be addressed through proofreading.

I suggest revision to respond and address lingering issues including the above and the line-by-line comments given below. An annotated version of the PDF file is also attached, with some language suggestions.

The authors appreciate the constructive comments and generally positive feedback from the referee. We hope that the missing information and the mentioned inconsistencies are now clear. In the following, we address the open questions and line-by-line comments.

**SPECIFIC COMMENTS**

l.2/3: one should not speculate about "jets" or "ramps" in the abstract without some evidence, or connection within the work or literature (as mentioned in the first review). Such a speculative "illustration" might be possible to include in the introduction, if citations are given to link to CWS.

In the previous revision, we mentioned that the concepts 'ramps or jets' were given for illustration, rather than for formal comparison. However, we accept that the connection of those phenomena to the CWS periods has not been proven. Therefore, we have now deleted the sentence from the manuscript.

l.5: as mentioned in the previous review.

A modification to the word 'extremes' has been done.

The following reply also covers the comment on l.158/Table 1.

The referee is correct. We have now emphasized the 'offshore' nature of the data considered for the analysis. Accordingly, we have modified l.6.

However, we want to point out that similar results on the behaviour of $p(T_c)$ have been observed when analyzing onshore data. Data from the Wettermast Hamburg have been investigated. The mast covers inland meteorological measurements up to 330m in height.

Fig. 1 shows the results of $p(T_c)$ from measurements of the wind speed at (a)50m height and (b)110m height. The representation of the $p(T_c)$ follows the same as in Figs. 3 and 4 in the manuscript. The periods $T_c$ are normalized by $T_{max}$ and the values of $\alpha$ for the power law are given in the legends.

[Figure]

Figure 1: Normalized probability density functions $p(T_c/T_{c,max})$ for data from the onshore Hamburg met mast at heights of (a) 50m, and (b)110m. The value $T_{c,max}$ for each data set is defined after a binning process as the center of a bin containing at least ten of the largest measured periods. The threshold $\varepsilon = A\,\sigma_u$ for the CSR is calculated with $A = 0.3$. The data in Fig. 1 correspond to measurements from January to December 2022. Similarly to the FINO data in the manuscript, only 10-minute periods with mean wind speed between 3 and 25m/s.

When comparing the onshore results in Fig. 1 to the results from the FINO data in the manuscript, it is clear that the value of the exponent $\alpha$ in $p(T_c) \propto T_c^{-\alpha}$, as well as the values of $\overline{T_c}$, $\sigma_{T_c}$ and $T_{c,max}$ are not universal. They depend on the terrain conditions and the assumptions for the calculation of $T_c$. However, the power-law behavior at the tails of the distributions $p(T_c)$ appears in both, offshore and onshore conditions. A statement on this is given in l.177 in the manuscript.

Deleted from the manuscript.

l.37-38: what does "The persistence phenomenon is straightforwardly recalled" mean? I'd suggest fusing it with the sentence it precedes, or remove.
We thank the referee for the suggestion. The sentence was merged with the one after it.

l.39: "inversely correlated" does not make sense alone; I presume you do not mean negative correlations, so this should be re-worded (perhaps without 'correlated'). It appears you are referring to persistence times being inversely proportional to some gust occurrence rates.
The referee is right, 'inversely correlated' might be misleading. With 'correlated' we meant 'associated' or 'related'. We thank the referee for the suggestion of 'inversely proportional to some gust occurrence rates'. Accordingly, we have modified the sentence in the manuscript.

l.158/Table 1: note these values might apply only for offshore since over land the surface-induced turbulence and flow accelerations are significantly stronger (consistent with the findings of e.g. Alcayaga 2017 [M.Sci.] or Kelly 2024). At any rate, it should at least be noted here (also probably in the abstract and conclusions) that the findings here are for the marine atmospheric boundary layer / offshore.
Please refer to the comment on l.7.

We thank the referee for the comment and the provided reference. A sentence and the referenced work have been added to the manuscript in l.155-158

Fig.3: this looks much better than the original, now with alpha∼=4. However, it is a bit confusing regarding the magnitudes. Could it be possible to plot all four heights/lines together, just with smaller dots? Or, just the small dots, since the straight lines are quite apparent?

We appreciate and have considered the suggestions from the referee for re-plotting Figs. 3 and 4 in the manuscript. However, we believe that in such a representation (i.e. without vertical shifting), the individual distributions $p(T_c)$ for different heights are difficult to distinguish and the figure looks chaotic.

On the other hand, vertical shifting is a common representation for individual PDFs in a unified plot. Unlike spectral densities, the magnitude of $p(T_c)$ is fixed by the normalization condition (i.e., $T_c, max$) and has no further meaning.

We decided to keep the vertical shifting in Figs. 3 and 4. Nevertheless, we have now decreased the width of the lines depicting the power laws $\propto T_c^{-\alpha}$ for better visualization. In addition, we have now added the '[a.u.]' to the label of the y-axis.

Fig.4: are the different $p(T_c)$ again shifted vertically? Suggest plotting them together as in Fig.3 suggestion, with much smaller dots (or just thin lines connecting the points).

Please refer to the previous comment.

l.192-195 / Fig.5: what $\sigma_u$, $U$ do these correspond to? Further: if using different CWS periods, one must normalize the data by respective $\sigma_u^2$ (or $u_*^2$ or $U^2$) in order to combine into one plot. Presumably, you did this, or did all the CWS have the same variance?

We thank the referee for the question. The information regarding the normalization of $u(t)$ for calculating the spectra was not given in the previous version. A sentence has been added to the text (l.191) and the caption of Fig. 5 in the manuscript.

Fig.5: Since a log-log plot is made to show the power-law, one cannot easily guess the variance "by eye"; however, perhaps to help display better the character you could plot f*E(f) to show the implied spectral peak.
We thank the referee for this important comment. This is certainly true for a more classical turbulence spectrum E(f), e.g, a von Karman or Kaimal spectrum that exhibits a deviation from $f^{-5/3}$ at small f, implying a spectral peak of $f*E(f)$. Nonetheless, due to the filtering of the CWS periods, Fig. 5 shows that E(f) solely possesses an inertial range $\propto f^{-5/3}$ and $f*E(f)$ would simply result in $\propto f^{-2/3}$.

l.198/§4.1: rather than the Kaimal spectral shape, it is the Gaussian prescription that is the problem; this should be directly stated (instead of just "Kaimal"). Also the heading should indicate your experimental data is from a wind tunnel. This reviewer suggests a subsection title like "Experimental wind-tunnel turbulence and synthesized IEC-standard Gaussian Kaimal"

We appreciate the suggestion from the referee and have adopted it in the manuscript. The title of the subsection has been modified. Moreover, the adjective 'Gaussian' has been added when referring to the Gaussian assumptions within the IEC-standard Kaimal model. Lines 198-202, 229, 233, 239 were modified.

l.226-234 / Fig.7 caption: how was 10s "chosen", not calculated for the synthesized dataset? This appears to be an arbitrary choice, and should be consistent with the Kaimal parameters you chose (Appendix E). It appears that 10s was chosen to make a better comparison, but is inconsistent with $L_{int}$=10m.

We thank the referee for realizing this inconsistency.

The referee is right, the parameters for generating the synthetic data were intended to be comparable to the atmospheric FINO data. That is certainly the case of the mean $\bar{u}$ and standard deviation $\sigma_u$ at the reference height (or 'Hub height') $H_H = 90$m. Therefore, $L_{int} = 170$m and $T_{int} = 17$s for the Kaimal

data. The calculation follows the definition of the Kaimal spectrum now given in Appendix E.

The value of $T_{int} = 10$s was incorrectly used for the normalization of the periods $T_c$ from the Kaimal data in Fig. 7 Therefore, the analysis was re-calculated with the correct value $T_{int} = 17$s. Fig. 7 is updated and the values in the text revised (lines 228-235). Note that the analysis and discussion are still valid after the correction from $T_{int} = 10$s to $T_{int} = 17$s.

l.298-299: note both the Kaimal/Veers and Mann models assume Gaussian-distributed spectral amplitudes.
We agree with the comment. The plural form 'IEC model**s**' has been adopted in the manuscript.

§4.2 / §5: it is nice that the CTRW model can replicate the alpha (i.e., tail of $p(T_c)$) found from the FINO3 data. However, as seen in Fig.9, it is quite sensitive to the Levy exponent; this should be mentioned. You could also remind the reader in the conclusion that the CTRW model was able to capture the statistics of rare (long) CWS events, when tuning $\alpha_{Levy}$.
This is a valid observation. l.275 and l.301 have been added to the manuscript to account for the suggestion.

Appendix A/l. : Fig.A.1 is a nice clarifying addition. However, to this reader, the blue and red lines in the bottom half do not appear to match what is shown in the top half (timeseries); the explanation does not seem to cover this either. Is there an error in the bottom half, or could this be explained better in the draft?
We thank the referee for this comment. We have revised Fig. A.1 and there is no error with respect to what we wanted to show. However, we agreed on a deficient explanation of the figure. Therefore we have improved it. We hope now it is clear, why the lines in the bottom half are not 'exactly' matching the crosses over the time series on the top half.

l.384-386/Appendix E: you should show the Kaimal form here, that you used.
We thank the referee for suggesting to describe here the Kaimal spectrum. By doing so, it became clear the calculation of the $L_{int}$ and $T_{int}$ values (aligned with the comment on l.226.).

l.384-385: setting the integral length scale to 10m is not consistent with your choice in §4.1 (see comments above around l.226-234/Fig.7) of $T_{int} = 10s$, unless U=1m/s.
Please refer to the explanation in the comment on l.226-234. This was a mistake. We have revised our calculations and the correct value is $L_{int} = 170$m. We again thank the referee for noticing.

**A new version of the manuscript is provided along with a diff file.**

**References**

Wettermast Hamburg. URL `https://wettermast.uni-hamburg.de/frame.php?doc=Home.htm`.